# Metallic Material Selection and Prospective Surface Treatments for Proton Exchange Membrane Fuel Cell Bipolar Plates—A Review

**DOI:** 10.3390/ma14102682

**Published:** 2021-05-20

**Authors:** Tereza Bohackova, Jakub Ludvik, Milan Kouril

**Affiliations:** Department of Metals and Corrosion Engineering, University of Chemistry and Technology Prague, Technická 5 Prague 6, 166 28 Prague, Czech Republic; jamborot@vscht.cz (T.B.); kourilm@vscht.cz (M.K.)

**Keywords:** fuel cell, bipolar plates, metals, coating, corrosion, interfacial contact resistance, United States Department of Energy (DOE)

## Abstract

The aim of this review is to summarize the possibilities of replacing graphite bipolar plates in fuel-cells. The review is mostly focused on metallic bipolar plates, which benefit from many properties required for fuel cells, viz. good mechanical properties, thermal and electrical conductivity, availability, and others. The main disadvantage of metals is that their corrosion resistance in the fuel-cell environment originates from the formation of a passive layer, which significantly increases interfacial contact resistance. Suitable coating systems prepared by a proper deposition method are eventually able to compensate for this disadvantage and make the replacement of graphite bipolar plates possible. This review compares coatings, materials, and deposition methods based on electrochemical measurements and contact resistance properties with respect to achieving appropriate parameters established by the DOE as objectives for 2020. An extraordinary number of studies have been performed, but only a minority of them provided promising results. One of these is the nanocrystalline β-Nb_2_N coating on AISI 430, prepared by the disproportionation reaction of Nb(IV) in molten salt, which satisfied the DOE 2020 objectives in terms of corrosion resistance and interfacial contact resistance. From other studies, TiN, CrN, NbC, TiC, or amorphous carbon-based coatings seem to be promising. This paper is novel in extracting important aspects for future studies and methods for testing the properties of metallic materials and factors affecting monitoring characteristics and parameters.

## 1. Principle of Fuel Cells

The proton exchange membrane fuel cell (sometimes also referred to as PEM/PEMFC) is an alternative to combustion engines. Despite this, fuel cells have plenty of advantages, such as a high energy efficiency [1,2,3,4], relatively low operating temperature (about 80 °C) [1,2,4,5], quick commissioning [2,5], low noise [2], and minimum exhaust emissions, which are important benefits for the environment [1,2,3,4,5]. However, there are limits to their widespread implementation, namely their limited lifetime, high acquisition costs, weight, and volume, which have to be reduced for use in vehicles [2,4]. The initial costs of the equipment are significantly influenced by the price of individual components—catalysts, membranes, electrodes, and bipolar plates [2]. The literature indicates that up to 90% of these costs are related to the production of bipolar plates and membrane electrode units [1].

The principle of a fuel cell is the conversion of a chemical energy into an electrical energy. In the case of PEMs, the fuel is a highly pure hydrogen and oxygen is the oxidizing agent. Hydrogen is fed into the anode where its bond is broken and H^+^ ions and electrons are formed. Protons travel through the membrane to the cathode where they react with the supplied oxygen and form liquid water; electrons go through an external electrical circuit and generate an electrical current (Figure 1) [1].

A fundamental component of a fuel cell is the membrane electrode assembly (MEA), which consists of a gas diffusion layer (GDL) and a catalyst layer [6,7]. Each MEA lies between two liquid-impermeable conducting plates; one of them serves as a cathode and the other as an anode [4,5]. In order to supply fuel and oxygen, distribution channels are made in these plates [5]. If there are two MEAs on both sides of a plate, one side serves as an anode and the other serves as cathode; this is called a bipolar plate. If there is a plate with only one function, then it is called an endplate [4]. The operating voltage of a single cell ranges from 0.6 to 0.7 V; however, most applications require higher voltages or greater power than a single fuel cell is able to produce, so individual cells are connected in series to larger units [7]. Cooling is ensured by inserting a cooling plate in the series of cells or the bipolar plates have their own cooling systems [4].

## 2. Bipolar Plates

Bipolar plates are one of the main components of fuel cells, and they affect cell performance and lifetime. They are irreplaceable due to many of their functions. They ensure the distribution of fuel and oxygen to electrodes and provide electrical connections between individual cells. They also serve as a mechanical support for MEAs and help to dissipate water and heat. To ensure all these functions, bipolar plates must combine excellent electrical and thermal conductivities, good mechanical properties, gas impermeability, and hydrophobicity [8] (the efficient drainage of excess water [9]).

Other equally important requirements for the properties of bipolar plates are given by the use of a fuel cell and the operating conditions [2]. For transport applications, the requirements are different from those of stationary applications [3]. For the former, bipolar plates, which represent about 80% of the weight and almost the entire volume of a fuel cell [3,4,5], must be light, thin, and be able withstand varying and demanding operation conditions [2,3,6]. Such conditions in fuel cells with a polymer electrolyte membrane correspond to environments with temperatures of 50–90 °C [10], high humidity, low pH, and varying loads [1,2,6]. For this reason, bipolar plates must satisfy the requirements of good corrosion resistance [3,5] and resistance to mechanical shock and vibration [2]. In the case of stationary units, on the other hand, the volume and weight parameters are not so important, durability, however, is more important [3]. For commercial use and mass production, price is the critical factor. The cost of bipolar plates is 30–40% of the price of a fuel cell [3,4]. It is therefore desirable, in terms of cost, weight, and volume, to produce bipolar plates of the smallest possible size and thickness from affordable materials and to use affordable manufacturing techniques [5].

Due to the large number of requirements for fuel cells, their components, and their various applications, limits have been established in the United States of America to ensure proper cell performance and lifetime; the so-called DOE limits (United States Department of Energy). These limits are always valid for a certain period of time and are revised regularly, depending on progress in fuel cell research. For the materials of bipolar plates, there are minimum values of electrical conductivity, flexural strength, corrosion resistance, and hydrogen permeability among the monitored varieties. The DOE requirements for PEMFC bipolar plates intended for transport applications are summarized in Table 1.

## 3. Materials for Bipolar Plates

Due to the many requirements for bipolar plates, many materials have been tested, because desired properties often inversely depend on each other. The original fuel cell used graphite for its chemical stability and had excellent electrical properties [14,15]. The main problem is its fragility, porosity, and the time and cost of production [1,2,14]. However, it is still used as a standard for other materials [16,17]. At present, research is focused on carbon–polymer composites, and on metallic materials and their surface treatments [2,4]. This review is focused on metallic bipolar plates, mainly stainless steel and its surface treatment. The review sums up various techniques for coating bipolar plates, consequential testing in simulated fuel cell environments, and compares results to bare materials, other coatings, or DOE 2020 targets, respectively.

### 3.1. Composites Materials

The polymer–carbon composite material consists of a polymer matrix and a filler. Different polymers are used as matrices and different forms of carbon are used as fillers [2,14]. Their advantages are good corrosion resistance, low weight, and lower costs of bipolar plate production compared to graphite [14,18], the disadvantages are still a relatively low conductivity and mechanical resistance in comparison with metals [18]. These properties are influenced by the type of matrix and filler, the size and shape of the particles, their distributions and amounts, and the orientation or method of manufacture [18,19]. Electrical conductivity can only be improved by increasing the filler content, but this is associated with a decrease in mechanical strength and the processability of the composite [2,4,20]. Achieving a balance between acceptable values of electrical conductivity and mechanical resistance is, therefore, the greatest challenge and these properties are also often studied in composite materials.

Both thermoplastics and thermosets can be used as polymer matrices. The advantages of thermosets are their higher strength, creep resistance, dimensional stability, and the possibility of obtaining a composite with a higher filler content due to a lower viscosity. On the other hand, they are less durable and their processing involves a curing process that is more time consuming. In addition, the process releases gases that can increase the porosity of a material [4]. Thermosets are tested for epoxy and phenolic resins, for thermoplastics polypropylene, polyvinylidene fluoride, polymethyl methacrylate, polyethylene, polyphenylene sulfate and polyetheretherketone, nylon or polyethylene terephthalate [4,18]. Various forms of carbon can be used as fillers, e.g., graphite, expandable graphite, carbon fibers or nanotubes, graphene, and carbon black [4,18,19]. The advantage of graphite is its higher electrical conductivity and the rugged morphology of the soot makes it possible to combine it with other fillers, where they fill holes [4].

### 3.2. Metallic Materials

The second group of materials tested for bipolar plates are metallic materials. Their indisputable advantages are good mechanical properties, as well as their electrical and thermal conductivities. Low gas permeability and the ability to simply obtain desired shapes as well as their low price are further advantages [1,2]. Thanks to their good mechanical properties, they can be used to produce thin metal plates, reducing the weight and volume of a fuel cell assembly, which reduces the disadvantage of their higher density [3]. The main difficulty with these materials is their corrosion resistance in the fuel cell environment, which is acidic and humid with temperatures around 80 °C. Most metallic materials do not withstand these conditions and corrode, which results in the release of transition metal ions into the fuel cell space. This is undesirable because these metallic ions contaminate the membrane and the catalyst layer and result in a reduction in cell performance and, in the worst case, failure. On the other hand, metallic materials that resist the environment are expensive and difficult to access (e.g., gold) or their corrosion resistance is based on the formation of a passive layer [4]. The formation of a passive layer occurs at the expense of electrical conductivity; an insulating layer is formed and the resistance between the bipolar plate and the diffusion layer increases as result of the so-called contact resistance. This is responsible for about 59% of fuel cell losses [2]. The greatest challenge for metallic materials is to find an inexpensive material with a good corrosion resistance and low contact resistance. The solution to this can be found by modifying the composition and thickness of a passive layer or by using conductive and corrosion resistant coatings, which are currently the most promising and most studied method [6].

## 4. Methods of Testing Properties of Metallic Materials

For the use of metallic materials as bipolar plates, the main goal is to achieve good corrosion resistance while not increasing the contact resistance; thus, these two properties are also the most often studied and compared with applicable DOE requirements for a given period.

### 4.1. Electrochemical Testing Methods

DOE requirements list two electrochemical methods for corrosion testing: potentiodynamic and potentiostatic testing. Potentiodynamic polarization is recommended for anodes ranging from −0.4 V to +0.6 V vs. ACLE (silver–silver chloride electrode) with a polarization rate of 0.1 mV∙s^−1^. The potentiostatic test at 0.6 V vs. ACLE for 24 h in aerated solution is intended for cathode. Both tests are performed in an electrolyte of pH = 3 with 0.1 ppm HF at 80 °C [11]. However, other methods, test conditions, and solutions can also be found in the literature.

A majority of authors use potentiodynamic polarization for rapid corrosion tests under anodic (nitrogen purging) and cathodic conditions (air or oxygen purging). Based on this, they evaluate the free corrosion potential, calculate the polarization resistance or determine the corrosion current density and the corrosion rate from the Tafel slopes. Other monitored properties are passivation ability, passive region width, or breakdown potential. Within potentials of −0.1 V and 0.6 V vs. SCE (saturated calomel electrode) potentiostatic measurements are made, which simulate the potential in anodic or cathodic fuel cell environments, respectively [21]. These tests run in different durations, from a few minutes to tens of hours. Short-term potentiostatic measurements in the range of 4–10 h predominate, although the DOE recommendation indicates 24 h. Short-term potentiostatic tests are only a rough estimation of the corrosion process in real PEM cells, which are expected to operate for 5000 h without damage [22]. Some authors apply higher voltages of 1–1.6 V vs. SHE (standard hydrogen electrode) in potentiostatic tests [16,23,24,25,26]. They report that automotive fuel cell operation will repeat processes, such as start or stop, with the potential temporarily reaching 1.4–1.6 V vs. SHE, while increasing the risk of fuel cell damage [24,27]. To a lesser extent, experiments based on cyclic voltammetry/polarization are performed [16,28,29]. One of these cyclic voltammetry methods is polarization according to ASTM G61, which is suitable for a first comparison of chosen materials [28]. The test starts with the stabilization of free corrosion potential for 1 h, followed by anodic polarization watith 0.6 V∙h^−1^ until the current density reaches 5 mA∙cm^−2^, then the polarization is reversed.

Electrochemical impedance spectroscopy (EIS) is applicable, especially for studying the electrical behavior of a coating or passive layer in a corrosive environment [10]. Capacitance and resistance properties related to layer thickness and resistance are monitored—the existence of defects, such as pores or cracks, means easier penetration of electrolyte through the coating or passive layer, which results in a decrease in the number of protective layers [10]. The most frequently selected parameters correspond to the frequency range of 100 kHz to 10 mHz at free corrosion potential with an amplitude of 5–10 mV [30,31,32,33]. Another possibility is to use the same frequency for measurements at potentials corresponding to fuel cell conditions, −0.1 V and +0.6 V vs. SCE [34,35]. EIS can also be performed in the potential range of 0.2 V to 0.7 V vs. SHE with a step of 50 mV, each spectrum being recorded between 0.1 Hz to 2 kHz with an amplitude of 14 mV [16].

A variety of published studies are on the exposure conditions of the monitored materials—especially the temperature, composition, and pH of the electrolyte. Corrosion media can be divided into two groups—test solutions simulating fuel cell conditions and solutions for accelerated tests [34]. Simulated solutions contain fewer anions with a pH closer to that of fuel cell operation and are more representative of actual conditions [34]. Their compositions are based on electrolyte analyses (ICP-AES, ICP-MS) after long-term operation of model fuel cells [16,28], and this corresponds to low-molar sulfuric and hydrochloric acid solutions with added fluoride ions, e.g., 12.5 ppm H_2_SO_4_ + 1.8 ppm HF or 0.01 mol∙L^−1^ HCl + 0.01 mol∙L^−1^ Na_2_SO_4_ [34]. The corrosive environment for accelerated testing has a much more acidic character, corresponding to different concentrations of sulfuric acid, with or without fluoride ions. Often the chosen medium is 0.5 mol∙L^−1^ or 1 mol∙L^−1^ sulfuric acid with 2 ppm HF [34].

Inconsistency in the choice of the electrolyte may be related to the differently reported aggressiveness of the fuel cell environment in the literature. In studies, it is possible to find information on the pH of a fuel cell environment in the pH ranges of 3–6 [34], 2–3 [1,36,37,38], 2–4 [30], or 3–5, depending on the extent of membrane degradation and hydrolysis of metals released from the bipolar plate [27]. Frequently, there are low pH values reported for the catalyst layer and Nafion membrane (a commercial name for copolymer of tetrafluorethylene and the perfluorovinyl ether with sulfonyl fluoride terminal groups). However, these are not directly in contact with the bipolar plate, and these values of pH might be inaccurate; moreover, it may be different for the anode and cathode [10,27]. Increasingly, under the influence of pH testing of the fuel cell environment, and under the influence of the DOE requirements, less concentrated solutions with a higher pH are tested, such as sulfuric acid with a pH = 3 [27]. Some authors also recommend a higher pH, e.g., Rajael simulated cathodic conditions with an electrolyte of pH = 5 for the reason that, at the beginning of fuel cell operation the pH = 2–4, but after a while, it will increase to 6–7 [39].

Inconsistent testing conditions are also supported by the choice of temperature. Although authors consistently report that fuel cell operation is associated with elevated temperatures of around 70–80 °C, many of them perform simulations only at laboratory temperatures [40,41,42,43,44]. The choice of temperature can, in principle, affect the course of the corrosion process; while at room temperature, the material can show satisfactory values, at 70 °C its passivation ability and the stability of the passive layer can be reduced or even disappear [45,46].

Electrochemical tests are performed in a three-electrode connection. Platinum and graphite are chosen to a lesser extent [21,47] as counter electrodes. In the case of the reference electrode, there is a concern about possible electrolyte contamination with chlorides [48,49], so some authors prefer mercury sulfate electrodes over the more common saturated calomel or silver-silver chloride electrodes. When using a saturated calomel electrode, contamination is avoided by using the Luggin capillary and salt bridges. Prior to the experiment, the passive layer and corrosion products were cleaned off, most often using mechanical abrasive papers and subsequent cleaning in an ultrasonic bath. Another option is electrochemical [10] or chemical cleaning [50]. Before measuring corrosion properties, samples are stabilized in an electrolyte, usually for 1 h. The complexity of the experimental setup depends on the number of parameters included in the experiment. This means that the single drop test is an undemanding method for measuring corrosion properties, requiring only a very small volume of electrolyte retained in an O-ring [51]. Rather, it is intended for rapid comparative measurements because it does not allow the introduction of gases; the tests are run at room temperature and the conditions are quite different from those in fuel cells. More complex electrochemical cells already use a heating bath [38] or introduce gas [23].

### 4.2. Fuel Cell Test

More realistic arrangements and operations of a fuel cell are reflected by the so-called single cell tests or fuel cell tests, in which a functional model cell or multi-cell connection is constructed in which the individual cells operate simultaneously under the same conditions [27]. Individual components for assembling a fuel cell include bipolar plates with a formed channel system, MEA, metal collectors, and endplates. Bipolar plates are most often tested with a “serpentine flow field” design [26,52,53,54,55,56]. An assembled cell of a commercial MEA, carbon fabric, and bipolar plates with single serpentine flow fields, clamped between two endplates, is shown in Figure 2 [26].

A fuel cell test is used to test the ability of a cell to provide performance as well as its long-term stability. To determine the maximum power of a model cell or the optimum conditions that ensure sufficient power without a significant voltage drop, I-V measurements and I-P characteristics are used (Figure 3) [21]. If a cell works properly, the values are stable over time. In addition to the operating temperature, other parameters, such as temperature and stoichiometry of the feed gases, their humidity (ensuring the presence of liquid water at the BPP/GDL/MEA interface [23]), the gas flow rate, and the operating pressure are tested. The time needed for these tests ranges from tens to hundreds of hours. Simpler tests are associated with the operation of model cells under constant conditions, e.g., at 75 °C, 0.5 A∙cm^−2^ current density, and ambient pressure [52] or at 80 °C, 0.7 V voltage and 0.3 MPa pressure (absolute) [57]. More complex tests already note that cyclic changes better suited to engine start-up, operation and shutdown processes, and related changes in humidity or temperature [27]. Cyclic changes in relative humidity increase fluoride ions leaching from MEAs [51]. An example of a humidity cycling test is an experimental set up between 40–100% humidification, with a temperature of 80 °C, and constant current density of 0.3 A∙cm^−2^ [51]. Another operates at 80 °C with a current density of 0.5 A∙cm^−2^ with a daily shutdown at which point the cell is cooled to 20 °C, which provides temperature and humidity cycling [58]. Another setup is to cycle the voltage values, e.g., 1.35 V and 1.8 V over twelve-hour periods, with a temperature of either 70 °C or 23 °C [59].

Corrosion and durability tests of fuel cells include analyses of ions leaching from bipolar plates (ICP-AES and ICP-MS) [60,61], MEAs, and GDL contamination after exposure expiration (SEM/EDS) [47,52]. A fuel cell for transport applications should be able to withstand 5000 h of operation without a significant decrease in power supply, while the number of ions released should not exceed 1 ppm/500 h [22,38].

Fuel cell tests are also used to characterize the aggressiveness of a fuel cell environment, as studies do not fully agree on pH and electrolyte composition. pH can be measured ex situ on the cathodic and anodic sides of a bipolar plate [10], or in situ using a commercially available combined micro pH electrode, inserted into the cell through small holes in the endplates [27]. In ex situ measurements, a decrease in pH at the anode from 5.6 to 3.4 was measured under operating conditions, 50 °C, and at a constant current density of 0.5 A∙cm^−2^. During in situ measurements (70 °C and 0.5 A∙cm^−2^) the pH at the cathode ranged from 3.4 to 4, and from 5 to 7 at the anode.

Interfacial contact resistance (ICR) or specific contact resistance is defined as the total resistance of an interfacial layer induced by a current passing through layers [62]. The contact resistance between the bipolar plate and the diffusion layer can be measured both in situ and ex situ, in principle. The latter is a simple and quick measurement to determine an approximate value, while in situ measurement offers more accurate readings when operating individual cells or connecting them in a series to a larger unit [63]. One in situ measurement variant consists of inserting a thin gold wire into an MEA during manufacture (Figure 4) and subsequently measuring the voltage drop between the wire and the bipolar plate during fuel cell operation (Figure 5) and the resistance calculated from it [64]. The more complicated arrangement consists of placing the gold wire in two places. The first is located between the GLD and the cathodic bipolar plate to which it is welded, the second is between two GLDs (Figure 6). The voltage is measured between the GLDs and the ICR is calculated from it. However, measurements are only made at the cathodic site, since measurement at both the cathodic and anodic sides is very difficult in this arrangement [65].

To evaluate ex situ contact resistance and its dependence on pressure, the procedure by Davis has been proposed in the past and modified by Wang, and has been used by most authors. The design of the experiment consists of placing a sample between two diffusion layers (carbon paper), which are inserted between two copper electrodes connected to an external current source (Figure 7). This system is then compressed at defined pressures at a constant current [63]. The current loads used range from tenths [31,36,60,66,67] to units of amperes [25,26,35,40,41,45,58,68]. However, a higher current can cause physical changes in the contact area at a microscopic level, e.g., due to local heating; therefore, Yoon recommended to avoid high currents and to measure properties at a current density maximum of 1.5 mA∙cm^−2^ [38]. From the measured voltage values, it is possible to calculate the total resistance, the sum corresponding to the partial interfacial contact resistance, and the resistance of materials (Equation (1)). As a reference measurement to calculate the contact resistance between the sample and the GDL, a sample-free system is measured under the same conditions and, similar to Equation (1), the total resistance is calculated according to Equation (2). Using the assumption that the resistance of the diffusion layer and the sample are small compared to the other values, they can be neglected and the contact resistance between the sample and the diffusion layer can be calculated from these two measurements, according to Equation (3) [63,69,70].
(1)$$R_{1}=\frac{U}{I}=2R_{Cu}+2R_{Cu/GDL}+2R_{GDL}+2R_{GDL/SAM}+R_{SAM}$$
(2)$$R_{2}=2R_{Cu}+2R_{Cu/GDL}+R_{GDL}$$
(3)$$R_{GDL/VZ}=0,5\left(R_{1}-R_{2}-R_{GDL}-R_{SAM}\right)\approx 0,5\left(R_{1}-R_{2}\right)$$
*R*_1_—total resistance, *R**_Cu_*—resistance of one copper plate, *R**_GDL_*—resistance of diffusion layer, *R**_Cu/GDL_*—contact resistance between copper plate and diffusion layer, *R**_SAM_*—resistance of tested material (sample), and *R**_GDL_*_/*SAM*_—contact resistance between diffusion layer and sample [69].

A typical representation of the contact resistance measurement is the dependence of ICR on the amount of contact force (Figure 8). As the contact resistance depends on the conduction contact diameter, the magnitude of the compaction force affects the contact surface and indirectly interfacial contact resistance as well [50]. Therefore, a pressure of 140 N/cm^2^, corresponding to the mechanical load in the practical fuel cell stack [48], was chosen within the DOE limits to compare different results. The surface area and hence the contact resistance can also be affected by surface roughness. The properties of a passive layer, such as its corrosion resistance, surface conductivity, or composition and thickness, are other parameters that may translate into ICR values [35,50]. In order to correctly evaluate the resistance properties of a material, it is necessary to measure contact resistances, not only at the beginning of the exposure, but also during short and long-term tests [67,71], because the characteristics of passive layer might change during tests [72].

In addition to contact resistance, some authors [51,73], when testing assembled fuel cell models, determined high frequency resistance (HFR), which corresponds to the total resistance of a cell, including contributions of MEAs and bipolar plates. This total resistance can be measured by electrochemical impedance spectroscopy or current interrupt, where contact resistance at the current density is measured before and after current interruption and HFR is calculated from their difference [51].

### 4.3. Surface Characterization Methods

Electrochemical tests and contact resistance measurements show the corrosion resistance and electrical properties of a material, but they do not say anything about the chemical nature of passive layers or coatings. The structure and morphology reflect a number of analytical and imaging methods, which are used to provide qualitative and quantitative information about a surface layer. The chemical composition is most often studied using Auger spectroscopy (AES) [25,51,74] and X-ray photoelectron spectroscopy (XPS) [16,53,75,76]. In the case of crystalline forms of coatings (nitrides, carbides) X-ray diffraction (XRD) can be used for phase composition [35,74,75]. For amorphous carbon coatings, where carbon bonds are studied, Raman spectroscopy is used [9,77,78,79]. Polymer-based coatings are characterized using infrared spectroscopy [80]. 

The morphology of a coating before and after exposure, its microstructure or its thickness can be monitored using electron microscopy (SEM and TEM), and using an EDS analyzer, the elemental composition or distribution of elements in a layer can be evaluated [21,39,60,79,81]. Surface roughness can be evaluated using profilometers [39] or atomic force microscopy (AFM) [9,21,35,77]. The concentration profile without the need of cross-section is made possible by GD-OES [35,47,75]. Hydrophobic properties are also tested by measuring the contact angle [9,39,78], where a larger contact angle means a more efficient water drainage [78]. To a limited extent, authors have performed microhardness measurements [39,75,79] or, in the case of multi-layer systems, scratch tests [61,79].

## 5. Achieved Results with Metallic Materials

### 5.1. Uncoated Steel

Steel is the most studied material in terms of graphite replacement. As a metallic material, it has the needed prerequisites for use in bipolar plates, such as electrical and thermal conductivities, mechanical properties, availability, and processability. A disadvantage, however, is its susceptibility to corrosion in the fuel cell operation conditions associated with the release of ions that may contaminate the polymer membrane. Of the many various types of steels, the most attention is given to austenitic stainless steels, namely AISI 316L, which is also suitable due to its low cost and availability [2,4]. Other austenitic stainless steels, AISI 316, 304, 310 or 904L, are studied to a lesser extent. Some authors have also tested ferritic stainless steels, such as AISI 436 and 446. Despite very different results being achieved for particular steels (Appendix A), which may be due to different test conditions, authors agree that these materials need to be coated to achieve the required DOE standards [51,82]. Table 2 summarizes the compositions of the tested steels according to ASTM.

Stainless steel AISI 310S has a better corrosion behavior compared to 304 in a solution of sulfuric acid with fluoride ions at 80 °C. In the case of AISI 304, uniform corrosion was observed on the cathodic side, whereas the most serious attack was observed at the gas outlet during a single cell test. The reason for this might be a localized decrease in pH near the gas outlet, caused by accumulation of fluoride and sulfate ions leached from the membrane. In these conditions, iron oxides from passive layer are more soluble. [52]. In most cases, when comparing AISI 304 and 316L, 316L is the better choice. The exception is comparison of the materials in 0.5 mol/dm3 sulfuric acid and 80 °C, where a lower contact resistance for steel 304 was achieved [63]. In environments with different ion contents and pH 3.5 at 60 °C, relatively low corrosion rates for AISI 316L, 904L, and 254SMO steels can be achieved under cathodic and anodic conditions without causing local corrosion [28]. By increasing the aggressiveness of the environment (0.5 mol∙dm^−3^ sulfuric acid at 70 °C) in 316L testing, there is pitting corrosion under cathodic and anodic conditions. The corrosion resistance of the steel is higher under the cathodic conditions of the fuel cell when the environment is purged with oxygen [22].

The humidity and temperature of an environment play a major role in the assessment of corrosion resistance. AISI 430 and 304L steels are comparable in dry conditions, but 304L steel had better results in wet single-cell test conditions. The susceptibility of steels to corrosion increases with increasing temperature [73].

AISI 904L steel outperforms AISI 316L steel, and has a wider passivity range and different passivation current density. This is probably due to a different composition of the passive layer, which is richer in chromium and nickel. However, both materials should avoid exposure to potentials above 1 V vs. SHE under cathodic conditions. Above this potential, steels enter transpassivity and an excessive release of transition metal cations into the environment occurs [16]. This is undesirable in terms of membrane contamination and depletion of the passive layer by chromium, which transfers into the solution as hexavalent ions. The passive layer largely consists of iron oxides, which provide less effective corrosion protection [83].

The nature of the passive layer depends on the composition of a material and the environment to which the material is exposed. Alloying elements, therefore, play a very important role in the formation of the passive layer and affect its properties. This is particularly the case of chromium, the foundation of the passive layer of stainless steels. Austenitic stainless steels also contain nickel, which contributes to the protective effects. AISI 904L, 254SMO, and 654SMO steels generally have excellent corrosion resistance, but on the other hand, show a significant increase in contact resistance after exposure due to the formation of a low conductive passive layer. AISI 201 steel, which contains a relatively high amount of manganese (about 7%), shows a low increase in contact resistance after exposure. The passive layer is more conductive, but it does not have sufficient corrosion resistance for the fuel-cell environment [50]. In more alloyed steels, addition of molybdenum also plays a role. Molybdenum is already incorporated into the passive layer in small amounts as an oxide and improves corrosion resistance [84]. By enriching the passive layer with molybdenum, passivation current density is also reduced, and the stability area of the passive layer is expanded. The protective layer may also be less defective [85]. On the other hand, its extensive content (6–7%) may contribute to the deterioration of contact resistance properties [50]. S32205 duplex steel tested in 0.5 M sulfuric acid with 2 ppm HF at 70 °C showed a positive effect of molybdenum on corrosion and contact resistance properties, although the contact resistance still exceeded the DOE 2020 limits by several times [76].

Using electrochemical treatment, it is possible to modify a passive layer without changing the steel composition. The surface layer can be enriched with chromium and nickel and depleted of iron at the same time. Iron oxides are more readily soluble, therefore they preferentially dissolve under anodic polarization, while the other components of the passive layer dissolve significantly more slowly, or not at all [86]. Thus, the corrosion resistance of stainless steels can be increased and the contact resistance reduced [49]. Another contribution of this method is the smoothing of the surface as a result of the electrochemical polishing effect. Such a pretreatment appears to be more beneficial for subsequent coatings than a mechanically polished surface [87].

### 5.2. Steel Surface Treatment

Stainless steels are a promising material for bipolar and endplates, but they do not have sufficient corrosion resistance without surface treatment and have a contact resistance that is too high [2]. Suitable coatings must provide sufficient electrical and thermal conductivities, good adhesion to metal [47,88], must be gas impermeable, stable, corrosion resistant, and have as few defects as possible [88]. Defects in a coating accelerate its degradation, act as localized attack initiators, and can also retain water and further reduce the lifetime of materials [4]. Precious metal coatings, such as gold, silver, or platinum, are appropriate in terms of desired properties, but not in terms of costs [77,88], therefore many other materials have been tested (Appendix A). These materials are generally divided into metals and carbon-based materials. Metal-based coatings include, in addition to precious metals, metal nitrides and carbides, non-metallic coatings are represented by conductive polymers, in particular or carbon-based coatings [89]. These coating can be prepared by a variety of processes, but, for the commercial sector, the technology must be achievable in terms of costs, reliability and affordability [3]. The methods being tested include various variable parameters of carburizing, nitriding, PVD and CVD technologies or electroplating [41,89]. Most of the work is focused on 316L steel surface treatment and coating as the most achievable commercial material, and to a lesser extent 304 steel with other types of coated steel rarely tested.

#### 5.2.1. Surface Treatments Based on Nitrides

Metal nitrides provide the necessary properties of bipolar plates due to their good electrical properties, hardness, and chemical stability [32,57,88]. The most tested nitrides are TiN, CrN, ZrN, and NbN [69,89]. Nitrides are mostly produced by PVD technology [47,69,89]. PVD deposits vapors of materials onto a substrate, resulting in coatings that are often harder and more resistant than electrodeposited coatings [1]. The difficulty is to prepare a defect-free structure using PVD technology, which limits its use [6,57]. Therefore, alternatives, such as magnetron sputtering, are tested [1]. Another possibility is thermal nitridation, an inexpensive and well-controlled method for producing defective-free coatings [57]. This method produces mixtures of nitrides and oxides, but the problem is the formation of chromium depleted areas, which are potentially corrosion hazardous [4,35]. Brady tested a high-temperature nitridation on ferritic steels with 20–23% chromium and 4% vanadium [51]. The high oxygen and nitrogen permeability of the steel was dealt with by surface pre-oxidation. The stability of the cell at 500 h in a single cell test at low contamination with released ions showed promising results.

A great deal of attention has been paid to TiN-based nitrides, but without clear and satisfactory results. Mainly due to defects in the coating, localized corrosion was observed [29,90,91]. The PVD TiN coating on 316L steel showed the lowest contact resistance when tested in 0.5 M sulfuric acid at 80 °C and compared to CrN and ZrN coating, on the other hand, it was the most susceptible to corrosion and the contact resistance grew significantly after short-term tests [89]. Additionally, TiN coating prepared by magnetron sputtering on 316L steel showed an increase in ICR during potentiostatic tests [23]. The choice of sputtering conditions (flow medium pressure and voltage) can influence whether a TiN or TiN/Ti_2_N-based coating is formed. The combined coating has finer grains, fewer defects, and a greater consistency with 304 steel. However, even this structure does not withstand prolonged exposure in 0.5 M sulfuric acid solution with 2 ppm HF and exhibits grain intergranular corrosion [41]. Plasma-enhanced atomic layer deposition (PEALD) is capable of creating ultrathin, defect-free and uniform TiN of thickness 25–67 nm on AISI 316L. It is even possible to achieve an amorphous interfacial layer, which improves corrosion resistance [92].

In the case of CrN-based coatings, the results are different. This is due to the stoichiometry of the formed compounds. PVD technology produces coatings combining Cr, Cr_2_N and CrN phases, where the Cr_2_N phase shows a lower corrosion resistance than the CrN phases [89], but is usually electrically conductive [69]. The content of the CrN phase can be influenced, e.g., by the rate of nitrogen flow during sputtering [93]. Similarly, in plasma nitriding, the selected temperature is an important parameter influencing composition. In the case of 304 steel, at higher temperatures of 520 °C, a thicker layer, susceptible to localized corrosion, results from the less resistant (α + CrN) phase, while at 420 °C a more resistant γN phase, also referred to as S-phase, forms [35,72]. Comparison of TiN and CrN coatings in 0.5 M NaCl, where CrN is better based on the lattice due to a less-defective structure. The more complex CrN_x_ coating has a finer structure that better eliminates porosity, oxygen diffusion, and uneven corrosion attack [90]. The Cr-N layer produced by plasma nitriding of Cr-electroplated AISI 304 is composed mainly of Cr_2_N. Due to the good electrical conductivity and corrosion resistance of Cr_2_N, ICR values met DOE targets, even after a 4-h potentiostatic test, with only a slight increase (from 4 to 4.5 mΩcm^2^) and met DOE requirements in the case of current densities [94]. A thin coating can benefit from high toughness and deformability, saving on deposition time, but has higher demands in terms of quality and process control. Following these points, a 50-nm-thin CrN was prepared on AISI 316L using reactive magnetron sputtering. AISI 316L pretreated by etching and coated with CrN met the DOE parameters, even after stamping (20% deformation) [95].

NbN is another important metal nitride that was tested in fuel cells. It was prepared using various plasma nitriding techniques. Plasma nitriding followed by thermoreactive deposition (TRD) makes it possible to prepare thin and compact layers because nitrogen migrates from the CrN phases and reacts with niobium; at the same time though, the formed NbN acts as a barrier against nitrogen diffusion. However, without subsequent deterioration, ferroniobium particles are retained in the coating, which are responsible for inferior corrosion properties [72]. Despite pickling, the value of the corrosion current density is only around the DOE limit and the contact resistance is 3–4 times higher than the values of only the nitride material [35]. Active screen plasma co-alloying (ASPA) can produce a coating that, depending on nitriding parameters, has either a duplex structure with a niobium-enriched outer layer and an internal S-phase (nitrogen-saturated austenite) or a structure without S-phase. The coating formed on 316 steel achieves a contact resistance that corresponds to the DOE 2020 limits; the disadvantage is an increase in current density to values higher than those of uncoated 316 steel [75]. A similar experiment with platinum instead of niobium gave better corrosion results and the DOE limits were fulfilled. However, with the exception of the single-cell test, corrosion tests were performed at room temperature [53]. A double-glow plasma alloying technique can also provide a defective-free NbN layer. The uniform layer on 304 steel showed good corrosion properties below the DOE 2020 limits and the ICR values increased to twice the value of 18–19 mΩcm^2^ after a four-hour potentiostatic test [71]. A nanocrystalline β-Nb_2_N layer was prepared on AISI 430 stainless steel via disproportionation reaction of Nb(IV) in molten salt (NaCl, KCl, NaF). The coating adhered well to the substrate, having a dense and fine grain structure, and thickness of 600 nm, meeting the DOE parameters in terms of current densities (0.1 µA∙cm^−2^) and ICR (5.2 mΩcm^2^ after 500 h potentiostatic polarization) [96].

One of the nitride coatings tested was also a TaNx-based coating on 316L prepared by magnetron sputtering. The proportions of tantalum and nitrogen in the protective coating and the morphology of the coating were influenced by the rate of nitrogen flow during sputtering. The nitrogen sputtering velocity was associated with its higher content in the final coating and the reduction of the ICR was close to the DOE boundary (11 mΩcm^2^ at 150N∙cm^−2^). Excellent corrosion properties were also observed, although long-term potentiostatic tests are lacking [74]. The TaNx coating prepared on 316L steel by high-performance pulsed magnetron sputtering provided a dense and defective-free structure, but with an increasing amount of nitrogen, obtained a columnar structure that was more susceptible to corrosion attack. Nitride coatings also showed a significant increase in contact resistance after corrosion tests, the resistance properties are even worse than those of uncoated steels [25].

Due to the high melting point of zirconium, its low vapor pressure, and its sensitivity to oxygen and carbon contamination, it is not easy to deposit ZrN coating by PVD technology compared to other nitrides [59], therefore, there is only a limited number of published studies. One exception is the deposition of ZrN coating on 316L steel using the double glow discharge plasma technique [66,97]. Under cathodic and anodic conditions, the current density dropped significantly below the DOE 2020 limit and the ICR increased negligibly after 5 h, from 7.4 to 9.2 and 8.5 mΩcm^2^, which still meets the DOE 2020 limits [97]. Another tested nitride was MoN on 304 steel [91], but, despite a significant reduction of corrosion and resistance properties, DOE limits were not reached. Moreover, contact resistance after 4 h of exposure in cathodic and anodic environment increased from the original 27 to 34 mΩcm^2^.

Improved properties of nitride layers can be achieved by a multi-layer system that reduces the probability of defects passing through the entire coating thickness, reduces porosity, and improves electrical conductivity [67]. A Ti interlayer improves corrosion resistance and coating adhesion. CVD is the best method to prepare such an interlayer, however PVD is also possible. That made using CVD is denser. Nitridation of the Ti layer on AISI 316L can result in a Ti-Fe-Ni interlayer with an outer layer of TiN. The thickness and the performance of this coating system depends on the time and temperature. Coatings thinner than 5 μm degraded during experiments [98]. Depending on the deposition time, different coating morphologies and layer thicknesses can be achieved. In the case of the two-layer AlN + TiN system, the coating deposited for the longest time showed the best results, yet compared to uncoated 316L steel, the corrosion current density did not decrease in potentiodynamic tests and the initially good contact resistance increased tenfold after 50 min of potentiostatic tests [67]. PAPVD-prepared TiAlN coating exhibited worse corrosion properties than TiN and CrN coatings [69]. A multilayer system based on titanium interlayer, Ti-Cr-N transition layer (improved adhesion between layers), and CrN top layer on 316L steel has demonstrated the possibility of using arc ion plating techniques to create a defect-free structure improving corrosion resistance and reaching the DOE 2020 limit for ICR [68]. However, contact resistance was not measured after a two-hour potentiostatic test and a long-term test was omitted. The coating was also prepared in the opposite way, incorporating titanium into the CrN structure and forming a Cr-CrN-CrTiN stack [36]. By incorporating titanium, the coating became finer and free of visible defects. However, by increasing titanium content in the top layer, grain coarseness may increase due to the formation of the TiN phase. Addition of titanium decreased the corrosion current density below the DOE limit. In the case of ICR, the effect was dependent on the titanium content, but the required DOE limits can be achieved by appropriate concentration selection. Combination of arc ion plating and magnetron sputtering techniques allowed the addition of the effects of molybdenum to the TiN-based coating structure to be tested. However, this did not provide the desired results. Corrosion tests in 0.5 M sulfuric acid with 2 ppm fluoride ions showed that the addition of molybdenum did not improve the properties, but did reduce the corrosion resistance [99]. Incorporation of oxygen into TiN results in a TiN_x_O_y_ layer with a combination of the great corrosion resistance of TiO_2_ and the electrical conductivity of TiN. Corrosion resistance is granted even at high voltages, however, ICR is above the DOE targets (19.8 mΩcm^2^ after long-term polarization) [100].

Improvement of the CrN coating properties can be achieved using a Ni-P-based interlayer. A comparison of the behavior of the magnetron-sputtered CrN coating and the combination of CrN with an electroless precluded interlayer on medium carbon steels showed that the use of the interlayer improved both the corrosion and contact resistance properties, and the DOE 2020 limits were fulfilled, although the potentiostatic tests were only short-term tests [32]. The effect of aluminum addition to the CrN coating structure was also tested [101]. Using closed unbalanced magnetron sputter ion plating (CFUBMSIP), defect-free dense coatings with Al content in the range of 1.86–21.34% were prepared. The incorporation of aluminum atoms reduced the CrN lattice parameter and refined the grains. Only a low aluminum content (1.86%) had a positive effect and lowered the ICR to 5.1 mΩcm^2^. With content above 10%, it had a negative effect. In contrast, the increasing aluminum content had no significant effect on the corrosion properties, the values were very similar, although they exhibited slightly higher current densities compared to the CrN layer.

A multilayer Ta/TaN system prepared by magnetron sputtering produced a defect-free and adhesive coating that exhibited corrosion behavior below the DOE threshold (0.028 µA∙cm^−2^) [90]. Although the ICR results did not meet the DOE 2020 limits, the values remained stable after a 12-h potentiostatic test and only slightly exceeded this limit (13 mΩcm^2^ after the corrosion test). Amorphous Al-25Cr-5Mo-40N with a content of nitrides (AlN, MoN, CrN, CrN_2_) and oxides (Al_2_O_3_, MoO_2_) was comparable, considering corrosion resistance, with Ta/TaN on AISI 316L. However, the ICR (around 33 mΩcm^2^) did not meet the DOE targets [102].

#### 5.2.2. Surface Treatment Based on Carbides

Like nitrides, carbides have suitable properties—electrical conductivity, hardness, and corrosion resistance [88]. It is possible to obtain a carbide-based coating by means of traditional high-temperature cementation or by means of modern methods at lower temperatures using plasma. In these processes, the temperature, time, or gas composition affect the resulting coating structure and its properties. A disadvantage of the high-temperature process is that it is susceptible to the formation of a chromium-depleted layer due to the precipitation of carbides at the grain boundaries, which leads to an increase in hardness but a reduction in corrosion resistance. The most versatile technique is plasma cementation, which improves surface properties and allows carbides to penetrate deeper into the steel matrix structure [3].

It is possible to create a uniform chromium carbide layer with a high cohesion to the substrate using thermo-diffusion chroming. A Cr-rich layer composed mainly of carbides and Fe-Cr solid solution is prepared on AISI 316L using the pack chromizing process at 1070 °C. During 4 h, potentiostatic polarization has a current density lower than 1 µA∙cm^−2^ (0.14–0.16 µA∙cm^−2^) and the ICR met the DOE standards, even after a 4-h test [103]. Post heat treatment might be a possibility to decrease precipitation of corrosion products and lower the Cr in the passive layer, according to corrosion test in 0.1 M HCl and at room temperature [104]. However, conventional high-temperature applications above 1000 °C (ensuring diffusion and reaction kinetics) may be associated with degradation of the mechanical properties of the matrix and coating distortion [105]; therefore, low-temperature application with steel surface pretreatment has been recently tested. The pretreatment of the surface by electric discharge machining or rolling increases the diffusivity of chromium [106]. Spark erosion machining influences the structure and composition of the recast layer, increases the carbon content, and improves the thickness of the diffusion layer during chrome plating. Crystalline defects reduce the activation energy and provide a driving force for the increase in the diffusion rate or chromium deposition, which allows a reduction of the chromium plating temperature [107]. Such a pretreatment of the surface of AISI 1020 (EN. 1.1151) [107] and AISI 1045 (EN 1.1201) [106] medium carbon steels increased coating corrosion resistance and decreased contact resistance, but all measurements were performed in 0.5 M sulfuric acid at 25 °C. Under these conditions, a contact resistance value of 11.8 mΩcm^2^ for AISI 1020, machined using electric discharge (EDM), was achieved at a pressure of 140 N∙cm^−2^, which is more than a triple the decrease when compared to chromium-plated steel without pretreatment. For AISI 1045 pretreated by means of rolling, a value of 5.9 mΩcm^2^ was obtained, while for the material treated with electric discharge, 9.8 mΩcm^2^ was obtained. The composition of the layers corresponded mostly to Cr/Fe carbides, and to a lesser extent to nitrides. A combination of shot peening and subsequent chromium plating of 316L steel at 900 °C produced a thicker chromium-enriched layer, predominantly composed of Cr_23_C_6_ and Cr_2_N, compared to chromium plating at 1100 °C without pretreatment [105]. This layer exhibited better corrosion properties and lower ICR values, although the cathodic corrosion current density of 6.5 μA∙cm^−2^ did not meet the DOE 2020 limits. In addition, the achieved ICR values were only close to the DOE 2020 limits (13 mΩcm^2^ at pressure 200 N∙cm^−2^).

Electroplating is another method that can be used to deposit Cr-C coating from a chromium ion solution. Its advantages are its low cost, ease of operation, speed of coating or small constraint on the shape or size of the part. The disadvantage of this method is in finding a suitable solution composition and deposition time to achieve the desired layer properties. Depending on the deposition time in the range of 10–60 min, layers with thicknesses in the range of 1.3–6.3 µm can be prepared on 304 steel, with corrosion properties deteriorating with increasing coating thickness, probably due to defects, mainly cracks, and the lower content of carbon [55]. The thinnest layer that met the corrosion limits of DOE 2020 showed the best corrosion properties, and, after long-term deposition, it showed no visible defects. Insufficient values were only for ICR, which increased from 19 mΩcm^2^ to 29 mΩcm^2^. Galvanization can also be combined with subsequent thermal diffusion in a vacuum furnace, which leads to an increase in chromium content in the surface layer [108].

Different chromium content in the outer carbide layer of the CrCx multilayer system can be achieved using the CFUBMSIP method [109]. On the basis of XRD analysis, it was found that the prepared layers consisted of metal carbides, amorphous carbon and metallic chromium. The effect of chromium content on the sp^2^/sp^3^ binding ratio, as well as on the amount of chromium deposition, was observed. As the chromium content increased, a greater portion was deposited as metal chromium and the carbide content decreased, and this had a negative effect on the ICR values and corrosion current density. The coating of Cr0.75C5, which showed the lowest ICR (1.4 mΩcm^2^ at 1.4 MPa) and the best corrosion properties (105.4 A∙cm^−2^ and 106.5 A∙cm^−2^ after 10 h of exposition under anodic and cathodic conditions, respectively) was evaluated as the best coating.

Niobium refractory carbide, which has excellent chemical stability and high conductivity, even higher than ZrC and TiC, was tested on 304 steel using plasma surface diffusion alloying [88]. The defect-free coating with a thickness of 6–7 µm met the DOE 2020 limits for corrosion and contact resistance, and did not show any corrosion attack, even after 10 h of exposure in 0.05 M sulfuric acid with 2 ppm fluoride ions at 70 °C. Molybdenum carbide prepared by magnetron sputtering in various thicknesses (hundreds of nanometers) on 316L steel has been proven to be a very promising candidate for future use. Very low current densities were measured in 0.5 sulfuric acid with 2 ppm HF at 70 °C, 0.23 μA∙cm^−2^ under anodic and 0.091 μA∙cm^−2^ under cathodic conditions. Contact resistance also met the DOE limits after exposure tests (6.5 mΩcm^2^) [110].

M_n+1_AX_n_ is a general formula for MAX-phase coating, where M = transition metal, A = element from group 13–14, X = carbon, and n =1–3. These coatings are based on ternary carbides, and have good corrosion resistance and thermal and electrical conductivity. Defect-free Ti_2_AlC with an ICR of 3.3 mΩcm^2^ was prepared using a mid-frequency magnetron sputtering PVD technique [111]. Ti_2_AlC is also possible to be obtained by a combination of ECR-CVD and PVD methods at certain temperatures [112]. Ti_3_AlC_2_ [113] and Ti_3_SiC_2_ [114] were prepared by direct current and pulse magnetron sputtering with subsequent heat-treatment of AISI 304. Ti_3_AlC_2_ had better corrosion and conductive properties, and both met DOE targets even after 24-h polarization test. The corrosion current density of Ti_3_SiC_2_ increased to 1.23 μAcm^−2^ after 10-h polarization.

#### 5.2.3. Surface Treatment Based on Carbon

Coating of steels to achieve durability of metal bipolar plates is possible with a conductive carbon film with a high sp^2^ bond content, which is responsible for the electrical conductivity of a coating [48]. In addition to electrical conductivity, the advantages of carbon films are hydrophobicity and its chemical inertness [9]. However, there are some difficulties in the deposition of the carbon layer, since there is little adhesion between the steel substrate and the layer [24,109]. This can be solved by using a chrome, titanium, or silicon interlayer, which also reduces the risk of localized corrosion originating in coating defects [26]. In addition, the interlayer increases the ratio on behalf of the conductive sp^2^ bonds at the expense of non-conductive sp^3^ [45]. On the other hand, the usage of an intermediate layer elevates the price of the process. Another way of improving the carbon layer adhesion is to dope the layer with elements like chromium [24].

A thin layer of amorphous carbon, 3 µm thick, was created using the CFUBMSIP method on 316L steel. This coating results in a significant decrease in corrosion rate to values close to the DOE limits and a contact resistance value of 5.2 mΩcm^2^ at a pressure of 200 N∙cm^−2^. However, 8 h of lasting exposure in cathodic and anodic environments resulted in three times higher values, probably due to changes in the bonding of carbon atoms [9]. This can be avoided using the ion beam deposition method, where the prepared layer on 316L steel had a predominantly granular texture with minimal amorphous carbon. In corrosion tests, coated steel showed a rapid stabilization in the corrosion process, and current density values were below the DOE limits and, after a 7-h potentiostatic test, the coating did not show any changes. In addition, ICR values were stable, 12 vs. 13 mΩcm^2^ at 150 N∙cm^−2^ [78]. A similar coating, with very small crystallites, was prepared using the plasma-assisted CVD method on 304 steel. The coated material showed a significant decrease in contact resistance. Compared to uncoated steel, it dropped more than tenfold and, depending on the conditions tested, the values were around the DOE 2020 limits, before and after potentiodynamic tests [77]. An amorphous carbon layer prepared by direct current magnetron sputtering technique on AISI 316L lowered the corrosion current density, meeting DOE requirements. ICR was also low and met the DOE targets, even after a 12-h polarization. However, analysis after exposure revealed spherical defects enriched with Cr oxides at the interface of the metal and the coating. This is a sign of degradation of the coating and could be related to the increase in ICR, from 4 to 2.9 mΩcm^2^ [115].

With CVD technology, carbon is deposited rather in the form of tubes or fibers, but the surface morphology can be influenced by different proportions of acetylene and hydrogen in the mixture. The optimal conditions for the fine globular structure is to obtain a gas ratio of 0.45, and with a higher ratio, a coarser-grained structure is formed, while at a lower ratio, the morphology becomes fibrous. After 100-h exposure at 40 °C, the fine-grained coating structure with the magnetron sputtered nickel interlayers showed good properties, but longer-term tests at more realistic temperatures have not been performed [116]. Nickel interlayers have also been used in CVD deposition on 304 steel, which catalyzed the formation of the graphene structure. Graphene also appears to be a promising form of carbon coating. Compared to uncoated steel, there was a five time decrease in corrosion rate and an improved contact resistance, but tests were performed in 3.5% NaCl solution at room temperature [117]. Carbon coating can be doped with niobium (CFUBMSIP), resulting in a higher sp^2^/sp^3^ ratio Conductive sp^2^ bonds support a decrease in ICR to 1.2 mΩcm^2^. The effect of niobium depends of the amount doped into the carbon layer. Coating with high niobium content did not meet the DOE targets considering corrosion current density. This could be due to formation of NbC, where a mixture of NbC, Nb and a-C induce galvanic electrode speeding up the corrosion rate [118]. A similar effect of galvanic corrosion was described in doping a-C layer with silver and chromium [119]. Only low amounts of doping elements improve properties without significant side effects. The best result, a low ICR after 24-h polarization test (2.1 mΩcm^2^) and corrosion current density meeting DOE limits, was achieved by co-doping (CFUBMSIP) with 3.03 at.% Ag and 5.43 at.% Cr. The influence of the precursor gas, carrier gas, and plasma power intensities of the plasma-enhanced chemical vapor deposition on resulting microstructure and properties of carbon coatings were observed for AISI 316 [120]. The results showed that better adhesion and corrosion resistivity was achieved with methane as the precursor gas rather than acetylene. Lower deposition power increased the content of conductive sp^2^ bonds in the coating, plus benefited from a higher hydrophobicity. Another study of initial parameters focused on the deposition pressure in the term of direct current plasma enhanced chemical vapor deposition [121]. High pressure results in defects in the C-H bonds and the ICR increases. A good pressure in the case of a benzene atmosphere is 8 Pa.

The ratio between conductive and non-conductive layers can be influenced by doping the carbon layer with chromium. Using chromed pulsed bias arc ion plating (PBAIP), layers with different chromium contents were prepared, with the best results being obtained for a coating of 0.23 at.% Cr. At this content, the DOE limits are met, after 8 h of exposure at 70 °C in 0.5 M sulfuric acid with 5 ppm fluoride ions, but the surface showed localized corrosion attack [45]. With a plasma blowpipe using the internal arc method, a higher thickness of C-Ni can be produced [122]. A thicker layer of 480 µm has better properties, even though a thinner layer has a significantly reduced corrosion and increased conductivity. Contact resistance values of 5.8 mΩcm^2^, resp. 5.6 mΩcm^2^ after corrosion tests, and corrosion rates on the electrodes of the order of 0.01 μA∙cm^−2^ corresponding to a thicker layer met the DOE 2020 limits. However, all corrosion tests were only short-term.

The use of an intermediate layer has the above-mentioned effect on the adhesion of the carbon layer and the ratio between conductive and non-conductive bonds. It may also have a positive effect on corrosion properties at higher potentials, which are manifested in fuel cells; for example at starting and when the amorphous carbon coating alone cannot sufficiently withstand the high voltage [24]. The coating with the Cr interlayer resists potentials of up to 1.2 V, while the titanium and niobium layers resist up to 1.6 V. All layers, however, exhibit an increase in contact resistance after exposure due to an increase in oxygen content. Only the coating with the Cr interlayer and after an exposure at 1.1 V still reached the DOE 2020 limits, the other values are two to three times higher [24].

#### 5.2.4. Combined Surface Treatments and Coatings

Combining the benefits of carbide and carbon layers on 316L steel is a method to reduce defects [61,123]. Comparing the properties of Cr-C, Cr-CrN, C-CrN-CrNC, and Cr-CrN-CrNC-C coatings, it was found that all layers improved the corrosion or wetting properties of the material; thus, the best corrosion, adhesion and contact resistance values were achieved for the system with most layers. The required DOE limits were achieved with these coatings, moreover, there was no change in ICR after 10-h exposure [61]. By selecting the gas emission parameters during CFUBMSIP deposition, it is possible to influence its composition and properties. While the resulting ICR values are related to the content of sp^2^ carbon bonds in the layers below the upper amorphous carbon, the corrosion resistance depends on the Cr_3_C_2_ content in the CrNC layer. Although coating with this system met the DOE limits, to further improve resistance it would be good to optimize parameters, such as sputtering current and time, to increase the sp^2^ and Cr_3_C_2_ content, reduce the chromium content, and ensure the long-term stability of the PEM cell [123]. Coating of C-Cr and C-Cr-N on 316L steel prepared using PBAIP were also compared. Slightly increasing nitrogen and chromium content improved resistance properties in contrast to the pure carbon coating. Only the C-Cr coating reached values that met the DOE 2020 limits (8.72 mΩcm^2^ at 1.5 MPa). Corrosion properties were also improved [124]. The multi-layer Zr-C/C system also has promising results, with ICR values of 3.63 mΩcm^2^ at 140 N∙cm^−2^ respectively 3.82 and 3.92 after a 10-h potentiostatic test. The corrosion current density under cathodic conditions reached 0.49 µA∙cm^−2^ at a cathodic potential of 0.6 V/SCE—however, XPS analyses indicated changes in the coating that were not reflected in short-term electrochemical tests [125].

In addition to single element interlayers, combined interlayers, such as Cr + CrC, can be used with amorphous carbon, which exhibits the best properties compared to both chromium and titanium interlayers [79]. At the nanoscale, a structure with a morphology that most effectively blocks defects is created. Furthermore, the hardness and other important mechanical properties increase by the formation of carbides [26,79]. The amorphous carbon coating with a Cr-C interlayer was tested, for example, on 304 steel (Figure 9). Corrosion tests met DOE 2020 limits, and ICR was slightly higher (16.65 mΩcm^2^ at 150 N∙cm^−2^) [26].

The systems with a combination of a carbon layer and a nitride-based layer are also being studied. Adding nitrogen into an amorphous carbon layer on AISI 316L, using CFUMBSIP, improves its corrosion and conductive properties, resulting in achieving the DOE targets. The compactness of the protective layer also benefits from the formation of a CN_x_ phase. The corrosion current density was 10–40 µA∙cm^−2^ at 1.6 V/SHE. Operating at high voltages could be a problem and should attract more attention in future research [126]. Cr/N co-doping of diamond-like carbon on AISI 316L resulted in a smoother surface with better electrical conductivity, compared to doping only with nitrogen. Cr positively affected the ratio of conducting bonds, and coating with 6.67% Cr showed the best results [127]. Multilayer systems may provide improved resistance if low deposition rates are applied by means of CFUBMSIP [81]. Depending on the duration of the exposure, layers with a high proportion of sp^2^ bonds and an amorphous structure were prepared using this method. The combination of nitride and sp^2^ bonds achieved a contact resistance value in the range of 2.6–2.9 mΩcm^2^ at 150 N∙m^−2^ depending on the layer thickness. These values were lower than those of a pure carbon coating and even lower than those for gold-plated steel. However, it is not clear from the article [81] when the contact resistance was measured; the corrosion tests themselves were rather short-term in nature and did not fully correspond to DOE parameters (e.g., 10 h of potentiostatic test). Chromium was used as an adhesive interlayer for the deposition of TiN [48]. The properties of this system were compared with a single-layer system and thin layer of gold. After a short-term test, the multilayer system and the 10 nm gold layer met the DOE 2020 limits.

#### 5.2.5. Polymer and Composite Coatings

Conductive polymers as other coatings for metal bipolar plates are not studied like nitrides or carbides [30]. Polypyrrole (PPY) and polyacrylonitrile (PANI) prepared by electrodeposition were mainly tested using polymeric materials [80]. They offer corrosion resistance without releasing hazardous metal ions, and they have sufficient electrical conductivity and a simple and economical production [128]. However, the question regards their long-term stability in terms of exposure to a fuel cell environment. Despite the increase in corrosion resistance of 304 steel coated with PPY, the coating rapidly degrades with increasing exposure time in solution [129]. It is therefore necessary to ensure both conductivity and corrosion resistance at the same time [128]. A positive effect of the polymer layer based on poly-p-phenylenediamine (PpPD) was observed for 316L steel when exposed to 0.1 M sulfuric acid. Better properties were achieved by depositing the coating from a more concentrated electrolyte, with the surface having a smoother relief and corrosion below the DOE 2020 limits (ICR was not measured) [130].

The conductivity of the polymer layer can be influenced by doping it with suitable substances. Inorganic acids, such as sulfuric or phosphoric acids, have been tested, but due to the redox reactions at exposure they have been reduced and have lowered the corrosion resistance [128]. The use of organic substances, such as benzenesulfonate, p-toluenesulfonate or camphorsulfonic acid (CSA), might be a possibility. A PPY layer doped with CSA on 316L steel showed a shift in the corrosion potential to more positive values, and a notable decrease in the corrosion current density to the DOE 2020 limits (0.00187 µA∙cm^−2^) and a low contact resistance (5.5 mΩcm^2^) [128]. Composite coating of PPY and graphene oxide had a lower number of defects, higher corrosion resistance and adhesion strength to substrate AISI 304. Composite coating withstood the 10-h polarization test, while pure PPY did not [131]. Formation of conductive polypyrrole–graphene oxide/polypyrrole-camphorsulfonic acid bilayer improved desired properties even more [132]. Increasing the conductivity of the polymer layer was also attempted by adding various TiN nanoparticles to the electrolyte to deposit in the PANI polymer coating [133]. The contact resistance improved from 367.5 mΩcm^2^ of the original PANI to 32.6 mΩcm^2^ for the nanoparticle coating. However, incorporating the particles into the coating requires proper adjustment of the conditions because the content of particles is responsible for decreasing the polymer deposition rate and forming thicker films. A TiO_2_ nanopowder layer doped with Nb showed better corrosion properties than bare AISI 316L and with a PANI coating, however, the experiment was only done at room temperature and the ICR was not measured [134].

The thickness of the coating has one of the biggest effects on the corrosion properties and contact resistance values. Repeated deposition can result in different thicknesses of polymeric coatings; however, three cycles are optimal for PPY and PANI to ensure sufficient adhesion with 304 steel, good corrosion resistance, and acceptable resistance properties. PPY is more conductive and achieves better contact resistance values, but PANI has better corrosion resistance [80]. The two different polymers can also be combined within polymer coatings, since a single-layer coating often contains defects, which is a way to create a localized corrosion attack [135]. An example is the formation of a bipolar bilayer with an outer PANI layer and an inner PPY layer [30,135]. The use of the bilayer reduces porosity and limits the access of the electrolyte to the surface of the substrate, and by using the doping layer, a system of two different ion-permeable films can be created. While the outer layer inhibits the migration of chloride ions from solution, the inner layer may block the transfer of metal ions [30]. The combination of PAMT (polymerized 2-amino-5-mercapto-1,3,4,-thiadiazole) with PPY is another double-layer coating system. PAMT is conductive layer with a comparable corrosion resistance to PPY/PANI. Creation of PAMT layers on PPY seems to be more promising than PPY on PAMT, based on corrosion tests at room temperature [136].

#### 5.2.6. Other Coatings

In addition to the already mentioned nitrides, carbides, carbon, and polymer coating, there are other coatings which are studied metals (Au, Ag, Ni, Ta, Nb, and Zr) [33,38,87,137], oxides (e.g., SnO_2_ and PbO_2_) [138,139,140,141], or various combinations, such as Ni-P, Ni-Mo [39]. Ni-Mo and Ni-Mo-P coatings prepared on 304 steel by electrodeposition form a more compact layer with the addition of phosphorus. They have nanocrystalline to amorphous character; however, as the deposition time increases, the molybdenum content in the layer decreases and its content also decreases from the presence of phosphorous. The susceptibility of 304 steel to localized corrosion is reduced by both coatings, and lower values were achieved by coating Ni-Mo, though they were six times the DOE 2020 limits. In terms of contact resistance, the Ni-Mo-P coating was better, and was also more stable in longer potentiostatic tests [39]. The microstructure and composition are able to be optimized through the pH of the electrolyte and the deposition current, as in the case of Ni-Mo-Cr-P coatings. More conductive passive oxide film can be obtained [142]. Unfortunately, coated AISI 1020 steel failed after 80 h of single-cell operation. The possibility of gold plating 304 and 310 steels requires a thickness of more than 10 nm to achieve sufficient corrosion and resistance properties [38]. Reducing the thickness of the gold layer while maintaining the corrosion and resistance properties has been achieved with polished 316L steel through nickel or titanium interlayers. Smooth and defective-free coatings have been created in which hydrophobicity and conductivity have been improved. No release of metal ions was observed by ICP-MS after a 10-h exposure to acidic environment pH = 3 and 80 °C [87]. Although Zr and ZrNb coatings on 304 steel meet DOE limits under anodic conditions; only the Zr coating was sufficiently corrosion resistant at the cathodic potential, but it showed a high contact resistance [38].

Due to the high resistance of lead oxide in sulfuric acid and its conductivity, electrolytic deposition of PbO_2_-based coating on 316L steel was tested [138]. The corrosion properties were improved, but localized corrosion due to the presence of pores can be a problem. Defects in the structure of the coating are also a problem for silver coatings; therefore, chemical passivation in chromate-free solution to improve the properties of the 316L steel coated with silver has been attempted [33]. A smoother coating was achieved with fewer defects and hydrophobicity, and the corrosion properties improved, but the corrosion current density was still well above the DOE limit. Passivation showed a slight increase in contact resistance, but still met the limit of 10 mΩcm^2^. Combination of metallic Ta with Ta_2_O_5_ prepared using CVD deposition on AISI 316L which performed better in terms of corrosion properties and had a low ICR (22.3–32.6 mΩcm^2^). Conductivity of this coating is higher than for Ta_2_O_5_ alone. It is dense and consists of carbon with Ta_2_O_5_ and α-Ta (bcc) crystals on surface [140]. Magnetron sputtering with a remote inductively coupled oxygen plasma (O-ICP) is another way to prepare an oxide-based layer on a metallic substrate. A multilayer system CrO^*^/Cr/304 steel had a lower corrosion current density compared to a naturally formed oxide layer (CrO_2_/Cr/SS), due to the denser and thicker surface layer. Electrical conductivity was also better represented by the minor increase in ICR after potentiostatic tests; however, it did not meet the DOE targets [141]. TiB_2_ is a potential protective coating due to its low electrical resistivity, good corrosion resistivity and hardness. A TiB_2_ layer prepared using a high energy micro-arc alloying technique (HEMAA) on an AISI 304 substrate resulted in better corrosion properties. The current density decreased by 3–4 orders of magnitude and the ICR was 19 mΩcm^2^ after 20 days corrosion test performed in 0.3 M H_2_SO_4_ + 2 ppm HF at 25 °C [143].

### 5.3. Titanium

Another studied metal material is titanium and its alloys. Its great advantage is its low density and corrosion resistance in highly acidic and humid environments [1,31,64,66]. In addition, titanium ions are less toxic to the catalyst and membrane than ions released from stainless steels [31], but the high cost is a limiting factor [6,144]. In addition to the cost of titanium alone, the need for coating is also limiting, since without coating there is a gradual increase in contact resistance due to the formation of a stable and non-conductive oxide layer [144]. Thus, in terms of costs, titanium as a material for PEM cell bipolar plates is more suitable for aerospace applications than for automobile transport [145].

Coatings, such as gold, amorphous carbon, IrO_2_ and platinum, have been tested on titanium. As with other metals, metal nitrides, carbides, and carbonitrides have also been tested (Appendix A) [31,66]. They improve corrosion resistance and contact resistance, but the characteristics are inadequate for PEMFC conditions, mainly due to coating defects and localized corrosion [31]. TiN is well compatible with titanium and corrosion tests have achieved a very low corrosion rate of 0.0086 µA∙cm^−2^ and the contact resistance meets the DOE limits, even after a 8-h potentiostatic test (4 mΩcm^2^ at 200 N∙cm^−2^); however, the tests were run at room temperature [146]. In another study of a coating prepared using the DC magnetron sputtering method, a contact resistance value of 7.2 mΩcm^2^ was achieved at 140 N∙cm^−2^ which is about seven times lower than that of uncoated titanium [147]. Powder immersion reaction assisted coating method (PIRAC) is a potential technique, alongside PVD and plasma methods, to prepare dense, well-adhered and protective layers based on TiN with a minor amount of TiN_x_O_y_ and TiO_2_. Deposition temperature influences chemical composition, microstructure and thickness of the layer. It shows good corrosion resistance with ICR meeting the DOE parameters. ICR after potentiostatic polarization is inconclusive in the term of comparing with other coatings, due to exposure only being done at room temperature for 12 h [148]. Good corrosion protection was shown by NbC and TiC coatings prepared using plasma surface modification. Despite a corrosion resistance improvement, ICR increased after polarization test, less for NbC (from 16.6 mΩcm^2^ to approximately 24 mΩcm^2^) [149], more for TiC coating (from 7.5 mΩcm^2^ to app. 18 mΩcm^2^) [150].

Considering titanium alloys, TiAl6V4 alloy with a nanostructured Ta_2_N layer was tested, at different temperatures and pH values [31]. An adhesive layer was achieved that met the DOE limits for corrosion properties, and was not significantly affected by either temperature or pH. However, the contact resistance was only measured after 1 h of exposure and was slightly higher than the DOE limits. A nanocrystalline ZrCN coating was tested on the same alloy. Although it was defect-free and adhesive, localized corrosion was observed after potentiodynamic tests. However, the contact resistance values were only slightly above the DOE limits (11 mΩcm^2^ before the test; 17 mΩcm^2^ after) [66]. A TiN coating prepared using the liquid phase plasma electrolytic nitridation technique [151] and a coating composed of TiN nanoparticles with an amorphous Si_3_N_4_ matrix was prepared using the double cathode glow discharge plasma technique and met the DOE targets in terms of corrosion current densities, but the ICR was slightly above DOE targets [152].

### 5.4. Aluminium

Aluminum as bipolar plate material is considered mainly due to its low production costs, low density, and material availability, but its corrosion resistance under fuel cell conditions and ion contamination are questionable [4]. When using the coating, it is possible to achieve DOE limits if an adhesive and defect-free layer is achieved [6].

A significant decrease in corrosion rate was measured after application of a PANI (polyaniline) coating to aluminum 6061. An orderly corrosion rate of 0.01 µA∙cm^−2^ was achieved, but potential degradation of the coating was not addressed during long-term exposure. In contrast, the PPY (polypyrrole) coating did not work at all in the same tests, probably due to defects in the coating [153]. With a polypropylene, carbon fiber and carbon black composite coating, a corrosion rate of 1 µA∙cm^−2^ can be achieved after 7 h of exposure to sulfuric acid (1 mol/L) with 2 ppm of fluoride ions and a reduced resistance to 21 mΩcm^2^; however, the coherency of the layer was recognized to be a weakness [154]. LaB_6_, TiSi_2_, TiC, TiB_2_ and CaB_6_ have been tested as various fillers for an ethylene-tetrafluoroethylene matrix composite coating, of which titanium carbide proved to be the best candidate based on corrosion tests. However, when testing a coating combining graphite and TiC as a filler, neither corrosion resistance (particularly under anodic fuel cell conditions) nor ICR limits were achieved due to defects in the coating [155].

Most of the authors who focused on aluminum suitability tested nitride-based coatings, especially CrN. Aluminum 5083 was coated using PVD with different thicknesses of CrN. This type of coating is not suitable if defects in the coating cannot be avoided. Porous layers were formed at lower thicknesses and micro-cracks at greater thicknesses of about 5 µm [47]. When comparing the CrN coating with the CrN/ZrN multilayer coating, CrN alone provided better results and the multilayer coating exhibited excessive brittleness [59]. Even Cr-C-Ni created through the thermal sputtering process failed to produce a completely defect-free coating that would meet DOE limits, although the corrosion rate and resistance values were of the same order as the DOE limits and were significantly better than uncoated 6061 aluminum [58]. Tests of TiN and CrN coatings and their combination with a carbon layer on 5052 alloy showed that the multilayer system exhibited better corrosion and resistance properties. The C/CrN combination was best evaluated with DOE limits for contact resistance (4.08 mΩcm^2^ at 150 N∙cm^−2^). However, the change in ICR values after exposure was not measured [156]. Using grid-assisted magnetron sputtering, it is possible to change the preferred orientation of nitrides and make a layer with a nitrogen gradient. TiN prepared using this method on AA 1100 performed better in terms of corrosion resistance compared to homogenous TiN layers and uncoated Al. However, values were already above DOE targets after a 1-h potentiostatic test [157].

Other possibilities to increase the corrosion resistance of aluminum are, for example, chromium carbide coating or conversion coatings based on Zr, Si, and Cr oxides. The zirconium-rich conversion coating on alloy 5052 achieved a corrosion rate of 5.76 µA∙cm^−2^ in a sulfuric acid solution (0.5 mol/L) at 80 °C, and no localized corrosion was observed after potentiodynamic tests [158]. Adhesion of conversion coatings can be improved by additives. Chitosan worked for SnO_2_ prepared on AA 6061, where a better adhesion was followed by corrosion resistance improvement [159]. Chromium carbide coated aluminum 6061 showed a reduction in corrosion rate, from 416 µA∙cm^−2^ to 65 µA∙cm^−2^, and contact resistance from 184 mΩcm^2^ to 15.5 mΩcm^2^ at 140 N∙cm^−2^ due to the conductivity of the Cr_3_C_2_ layer [58].

The influence of alloying elements, especially copper, on the corrosion properties of aluminum alloys and on the subsequently coated alloys has been also studied. Al 7075 alloy with 2% copper content and CrN-TiN coating performed worse in terms of corrosion and resistance properties than pure aluminum with the same coating [21]. A negative effect of copper on corrosion properties was also observed for pre-galvanized aluminum alloys, Ni-Co-P [46] and Ni-P coated aluminum alloys [160], where higher phosphorus content is desirable in order to achieve higher amorphous structure and better corrosion properties. In addition, repeated zinc plating makes it possible to achieve a better morphology of the N-P coating on galvanized aluminum 6061 and a lower corrosion rate of 4.4 µA∙cm^−2^, but the resistance properties were not measured [161]. Ni-P-Cr was prepared on AA 7075-T6 and consisted of 20 μm Ni-P interlayer with 10 μm of electroplated Cr. Ni-P alone improved corrosion properties and ICR came close to the DOE parameters (15 mΩcm^2^). Adding a chromium outer-layer improved corrosion properties even more, but increased the ICR enormously to 629 mΩcm^2^ due to the formation of electrical resistive chromium oxides [162].

### 5.5. Other Metals

To a limited extent, other metallic materials, with or without coatings—nickel, copper and magnesium alloys—have been also studied (Appendix A. Nickel alloys exhibited higher corrosion resistance and lower contact resistance in fuel cell environment compared to steels [163], but as with titanium, they are more expensive materials [6]. The results of their tests were published mainly before 2010. The comparison of candidate nickel alloys for water electrolyzer bipolar plates (Hastelloy C-276, Inconel 625, Incoloy 825) in various environments, at temperatures of 30–120 °C, concluded that the most suitable material for high temperatures and for environments with H_3_PO_4_ doped membrane was Inconel 625. The properties matched with the tantalum coated AISI 316L steel, while the corrosion resistance of titanium exposed to the same conditions was the weakest. In the same work, a positive effect of molybdenum addition on the corrosion properties of nickel alloys was observed [163]. Crystalline TiN was prepared via cathodic arc evaporation on Monel alloy with various thicknesses and roughnesses, and uncoated Monel alloy has poor corrosion and contact resistances. Coating with thickness of 1.5 μm performed the best properties, ICR of 8.5 mΩcm^2^ and corrosion current density of 0.53 µA∙cm^−2^ meeting DOE targets [164]. It is possible to prepare PPy coating on nickel alloys using galvanostatic electrodeposition. The coating has a cauliflower morphology with good adhesion to the substrate and is homogenous [165]. Nitridation of the electroplated Cr layers on Ni resulted in CrN with a thickness that was dependent on the time of nitridation. A thinner layer composed mainly of amorphous CrN showed better properties, a lower corrosion and passivation current densities, due to less defects [166]. Electrodeposition of Ni-C composite layer is a promising way how to improve corrosion properties of Ni and other metals [167]. Ni-Cr50 was evaluated the best of Ni-X alloys (X = Cr, Nb, Ti, V) based on polarization tests, which was further nitrided. Thermal nitriding of the surface resulted in further reduction of the ICR and its stabilization during a 4100-h test in sulfuric acid at pH = 3, temperature of 80 °C and 2 ppm of fluoride ions. The corrosion rate reached the DOE limits, up to 0.9 V. However, even the uncoated material had a lower and more stable contact resistance than the AISI 316L steel. The better properties of the nitride alloy were attributed to the multilayer coating structure that could not be prepared on AISI 316L steel [57].

Some authors [168,169,170,171,172] have also dealt with amorphous metal glasses considering the fact that they are not limited by the solubility of the dopant in the matrix, which makes it possible to achieve a high content of passivable elements in the material. As a result, alloys with excellent corrosion properties without defects and grain boundaries were designed. The Ni65Cr15P16B4 alloy and the metal glasses Ni-Nb-Ti-Zr or Ni-Cr-P-B showed lower corrosion rates compared to AISI 316L steel, but the contact resistance was slightly higher (54 vs. 50 mΩcm^2^ at 140 N∙cm^−2^) and did not meet the DOE limits [168].

Copper exhibited good corrosion, electrical and thermal properties, but it is too heavy for fuel-cell needs. Some of its lighter alloys, such as the copper-beryllium alloy C-17200, have been studied [173]. Although author of that study referred to the alloy as a potential material for reducing fuel cell losses, corrosion tests did not meet DOE limits. The advantage of copper alloy with 5.3% Cr could be found in the possibility of nitriding due to the segregation of chromium on the surface at the annealing temperature as a result of its low mutual solubility [174]. However, DOE limits were not reached in corrosion tests and the alloy failed under cathodic conditions at which point the copper was oxidized in the positions of the surface layer defects. On the other hand, before and after the 5-h potentiostatic test, there was no significant increase in contact resistance and the values for nitrided material reached DOE limits. In addition to the nitrided alloy, uncoated material also met the limits. A more recent study on copper alloys focused on a graphene copper coating that showed a beneficial effect on cell performance [175]. Although the synthesized graphene prepared by the CVD method has been presented described as effective and financially acceptable, more corrosion tests and contact resistance measurements are lacking. Composite coating consists of graphene, TiO_2_ and octadecylamine prepared by electrophoretic deposition technique on copper lowered corrosion current densities during potentiodynamic tests by several orders of magnitude [176]. ICR met the DOE targets, however, only before potentiostatic polarization because there were no such measurement.

The main advantage of magnesium and its alloys is their low weight desirable in aviation; on the other hand, they are highly susceptible to corrosion and have to be coated. In order to improve the surface resistance, the PVD method was used to deposit a single layer of metal nitrides or a multilayer system such as AlN/AlN+TiN on AZ31hp and AZ91 alloys [177,178,179]. Despite the reduction in corrosion rate, coating defects and resulting localized corrosion occurred. Multilayer Cr/CCrx/C coating prepared using PVD technology on the nickel-plated alloy GW83 was also tested [60]. The nickel interlayer was tested on the basis of a positive influence on the corrosion resistance of steels. The properties were compared with an alloy coated with carbon, nickel and Ni+C. The best result was achieved by a multilayer coating that met the DOE limits for contact resistance (2.97 mΩcm^2^ at 150 N∙cm^−2^) and current density. However, after potentiostatic tests, contact resistance increased significantly (15.6 mΩcm^2^ for cathodic and 52.6 mΩcm^2^ for anodic environment) and localized corrosion was observed. A simple carbon coating did not provide sufficient protection because the layer delaminates. In contrast, the nickel layer itself provides greater adhesion and hardness of the layer and provides partial cathodic protection [60,180]. Combination of polymers, carbon and metals is also a possibility, for example, PTFE/carbon cloth/Ag coating [181]. PTFE serves as a corrosion barrier, carbon cloth layer is an electrical conductor and Ag paste is adhesive layer. With this system, the current density dropped to a tenth of the value for bare magnesium (0.02 µA∙cm^−2^) after the potentiostatic test at the cathodic potential resulting in meeting DOE parameters. Despite the decrease in ICR from 100.6 mΩcm^2^ to 27.28 mΩcm^2^, it does not meet the DOE parameters.

## 6. Factors Affecting the Monitoring Characteristics and Parameters of the Bipolar Plate Candidate Materials

Composition and structure of a passive layer or coating primarily determine the electrical and corrosion properties of a bipolar plate material. These can be influenced by a number of factors. The characteristics obtained from experimental testing depend on the selection of test conditions, such as electrolyte composition and pH, temperature or applied potential. Different experimental conditions, thus, become a source of inaccuracies when comparing materials.

### 6.1. Composition and pH of the Fuel Cell Environment

Test solutions simulating fuel cell conditions and accelerated test solutions are two basic groups of corrosive media. Low-molar sulfuric and hydrochloric acid solutions with fluoride ion additions are used for simulated solutions. The medium used for accelerated tests is more acidic, especially sulfuric acid of concentrations 0.5 or 1 mol∙dm^−3^ [34]. Based on a set of tests with various corrosion media on 316L steel, Feng stated that in simulated environment, more positive open-circuit potential and lower passivation current densities are achieved. It was also found that in the cathodic environment of simulated solutions, passivation at lower pH is easier, and in accelerated test solutions for more concentrated sulfuric acid [34].

The influence of pH of electrolyte on the corrosion properties of 310S steel was discussed by Kumagai [52]. In 0.05 M sulfuric acid with 2 ppm fluoride ions, in which the pH was adjusted to 1.2–5.5 by the addition of sodium sulfate, he observed changes in corrosion behavior and passive layer composition. With increasing pH values the open-circuit potential of the steel shifted in the negative direction and the passivation current density decreased, thus the material was more corrosion resistant (Figure 10). When analyzing the composition of the passive layer, he subsequently found that, at a low pH of 3.3, a thinner layer rich in chromium oxides was formed, while at a higher pH the layer was thicker and enriched for iron oxides. Corrosion current increases in the case of 304 steel due to lower pH [10]. Changes in pH, respectively electrolyte concentration may also be accompanied by changes in the form of corrosive attack, while exposure 316L steel in 0.5 M sulfuric acid showed uniform corrosion, and localized forms were already recorded in 1 M sulfuric acid [34]. Higher current densities and a negative potential shift at a lower pH are, not only apparent in steels, but they are also found in TiAl6V4 titanium alloy exposed to 0.05 M sulfuric acid with 2 ppm HF and pH adjusted to 1.5–3.5 (Figure 11 and Figure 12) [31]. However, testing materials in an environment with a pH lower than 3.5 is not justified by Orsi, because there are no such acidic conditions in fuel cells [23]. This claim is supported by the results of pH measurements, where despite the different configurations and conditions of the experiment, the lowest value of pH = 3.2 was determined [10,16,27,28].

Just as the aggressiveness of the environment affects the corrosion properties, it also affects the contact resistance values, but in an opposite way. ICR increases with rising pH and decreasing temperature (Figure 13). As the passive layer dissolves more slowly in the environment with a higher pH or lower temperature, the passive layer becomes thicker and less conductive [31,52,182].

The presence and amount of fluoride ions produced by the degradation of the polymer membrane in the fuel cell environment is also a question of the composition of the corrosive environment. The extent of their influence depends greatly on the tested material, respectively surface treatment. When comparing the corrosion aggressiveness of the environment with and without 2 ppm of fluoride ions on the S304 steel, the presence of fluoride ions was visible in a narrower passivation area and the material was more susceptible to corrosion attack. When steel was carbon-treated, the effect of fluoride ions was not evident (Figure 14) [77]. The effect of fluoride ions addition to the corrosion medium on the corrosion resistance of Ti-Al6-V4 titanium alloy with or without surface treatment is shown in Figure 15. Increasing the concentration of fluoride ions in the range of 2–6 ppm leads to a shift in the open-circuit potential in the negative direction and an increase in the corrosion current density, which is associated with increasing the ionic strength of the solution by fluoride addition and lowering pH, although the effect is lesser on surface coated with ZrCN [66]. Comparing the effect of fluoride ions and oxygen supply in 0.5 M H_2_SO_4_ at 80 °C, the fluoride concentration in solution proves to be a more important parameter than oxidant facilitating the formation of passive layer [77].

### 6.2. Temperature

Although polymer membrane fuel cells operate at elevated temperatures of about 80 °C, many experiments are done only under laboratory conditions or with only slightly elevated temperatures. With rising temperature, the corrosion process is accelerated and the probability of defects in the passive layer increases [31]. Comparing the corrosion resistance of the Ni-Co-P coating on aluminum alloys at 25 °C and 70 °C, it appears that while the coatings are effective at room temperature and the substrate is protected, its protective function disappears when the temperature rises. Open-circuit potential shifts to more negative values and the corrosion current densities are in the order of tenths of mA∙cm^−2^ thus, even 100 times higher than the current densities at lower temperature [46]. A similar trend, the shift of open-circuit potential and the corrosion current density, can also be observed on the Ti-Al6-V4 alloy [31] or steel 316 (Figure 16) [183]. The significant effect of temperature is demonstrated by the results of corrosion tests on various types of steels and nickel alloys in the work of Nikiforov (Table 3) [163], although he used 85% phosphoric acid as the electrolyte, so the values cannot be compared with other results from environments based on sulfuric acid.

The opposite trend to the corrosion resistance at elevated temperature is shown by the results of contact resistance, which is positively influenced by a higher temperature. Since higher temperatures are associated with an increase in corrosion activity, thinner passive layers are formed, and the material exhibits a lower contact resistance [31].

### 6.3. Applied Potential

In polymer membrane fuel cells, it is possible to temporarily achieve conditions that correspond to a greater potential load of the cell of about 1.4–1.6 V. These conditions affect operation and service life [23,24,27]. High cathodic potentials can degrade the membrane, causing carbon corrosion in the diffusion layer or dissolving the catalyst, which can precipitate elsewhere as the potential decreases [184], disrupting amorphous carbon-based coatings, or affecting the bipolar plate itself [24]. High potential can change the composition of the passive layer and accelerate the corrosion process, which results in higher amounts of released ions (Figure 17) [16] or values of corrosion current density (Figure 18 and Figure 19) [23,24].

Exposure of 316L and 904L steels to cathodic potential above 1 V/SHE leads to transition from passivity to transpassivity [16]. In transpassivity, preferential dissolution of hexavalent chromium ions occurs and the surface layer is enriched with iron oxides, which are less resistant to an acidic environment [83]. The TiN coating is passivated in the range of 0.5–0.9 V/SHE by the formation of a protective oxide/oxynitride layer, but at higher potentials of 1.1–1.5 V/SHE, hydroxides may be formed, which are more soluble and thus less resistant [23]. The effect of the potential is also reflected in the contact resistance—higher potential means greater resistance for both uncoated 316L stainless steel (Figure 20) [185] and for nitride or amorphous carbon coatings [23,24].

### 6.4. Exposition Time

Short-term tests used for material evaluation take usually tens of minutes or hours, but these exposure times are too short for reaching a steady-state corrosion process and to correctly evaluate the corrosion and resistance properties. The contact resistance depends on the character of the passive layer, its conductivity and its thickness. Longer exposure of steel enriches the passive layer with chromium due to lower iron stability under given fuel cell conditions, resulting in higher corrosion resistance but also higher contact resistance [186]. The contact resistance results for coated TiN steels show that there can be many differences between 15 min and 24 h in terms of potentiostatic polarization (Figure 21) [23]. Although the greatest increase is achieved in the first hours of exposure, when the passive layer is formed most rapidly and the ICR value grows steadily.

The results of the polarization resistance measured by means of the EIS method also showed that the corrosion resistance of materials in the first hours of exposure may significantly differ and the information is not relevant. For stainless steel 316L or 904L, values stabilized after tens of hours (Table 4) [28]. The influence of exposure time on the corrosion resistance of the material is illustrated by graphs of current density values during potentiostatic test (Figure 18 and Figure 19), where the current density stabilization rate occurs depending on the material, surface finish, and the experimental conditions [23,24].

### 6.5. Surface Pretreatment and Choice of Manufacturing Process

In addition to setting the experiment conditions mentioned above, other issues related to sample production, or surface pretreatment may affect test results. Using electrolytic polishing compared to mechanical grinding, it is possible to achieve a smoother surface, which can affect the subsequent coating of steel. The coating deposited on electrolytically polished steel 316 is also smoother, resulting in a lower contact resistance [87]. By comparing chemical and electrochemical polishing of Fe-Ni-Cr it was observed that, regardless of the chosen procedure, surface pretreatment had a positive effect on the corrosion and resistance properties, with the lowest ICR values achieved after chemical etching [44]. On the other hand, chemically etched samples were susceptible to localized corrosion and electrolytically purified samples were not. When studying various parameters of electrolytic polishing, such as process time, solution composition or current density applied to various types of steels and nickel alloys, it was found that the effect of the process and individual parameters cannot be generalized, it depends greatly on the chosen material. While 316L and 316Ti steels exhibited better corrosion properties in the untreated state, electrolytic polishing for others meant an improvement of up to 50%. From untreated materials, 316L steel performed best. An undesired result was an increase in electrical resistance for all materials [43].

Manufacturing process parameters can improve corrosion and electrical properties. Stamped samples have lower contact resistance values in comparison with those of hydroforming process, larger channel dimensions have a negative effect [63,89]. A channel depth affects fuel cell performance when the same power was achieved for 304L and 430 steels at different channel depths [73]. However, when the coating is applied to steel, these effects appear to be negligible as the coating process and coating thickness have the major role [89].

## 7. Conclusions

Ongoing research shows that the replacement of graphite bipolar plates is not an easy task. The effort targets in achieving a combination of good corrosion resistance combining and good electrical conductivity. As a result, several materials or material combinations showed promising future.

Metals are promising materials to be used in the production of bipolar plates. The advantages of metallic materials besides others are mechanical properties, good electrical and thermal conductivity, they are easy to process. The major limitation is the fuel cell environment, which means for affordable materials corrosion resistance due to the formation of a protective passive layer, but this is at the expense of increase of contact resistance values. Titanium might be promising material which provides great corrosion resistance in a fuel cell environment, but it shows unacceptable contact resistance. The higher price of titanium might result in usage mainly in aviation. Stainless steel as a cheaper choice providing also good corrosion resistance but high contact resistance. This problem is supposed to be solved by appropriate surface treatment, coating or coating system. Regardless of a deposition method or coating material, it raises the price of bipolar plates and even fuel cells. Requirements for suitable coating are mainly electrical conductivity, corrosion resistance, dense and defect-free structure. Such a coating has not been prepared by conventional PVD and CVD technologies. Defect-free and durable coatings could result in lower demands on base materials. An increase of price could be eliminated in the future, if the deposition methods become widely available.

Nitrides, carbides, metal oxides, polymers, composites, metallic coatings and their mixtures are considered in research. They are prepared by various methods and preparation techniques to compare their properties and suitability for fuel cells. Some multilayer coating systems have already shown convenient properties regardless of the substrate material.

Enormous numbers of various materials and deposition make a comparison of results complicated. There are no global guidelines and regulations for testing procedures. For a basic assessment, it is advised to follow the procedures and conditions proposed by the DOE. These tests are simple, time saving, instrument-friendly, operated under reasonable conditions and also offer the possibility to compare materials with each other. However, when evaluating materials based on the test parameters proposed by the DOE, it must be kept in mind that these conditions are only indicative and do not fully correspond to the fuel cell operation conditions, particularly from the short-term duration point of view. Potentiostatic tests at higher potentials to which PEM cells are exposed, especially during commissioning or shutdown, might be a suitable addition to the recommended tests. The selection of electrode potential levels can play an important role. To simulate more realistic the operating conditions of PEM fuel cells and long-term testing, in situ fuel cell tests are required for promising materials.

## Figures and Tables

**Figure 1 materials-14-02682-f001:**
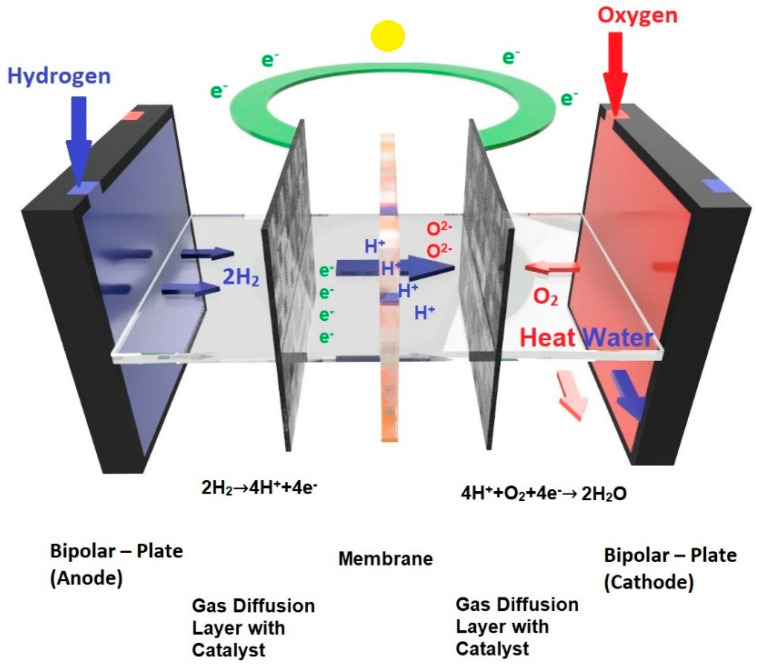
Proton exchange membrane fuel cell model.

**Figure 2 materials-14-02682-f002:**
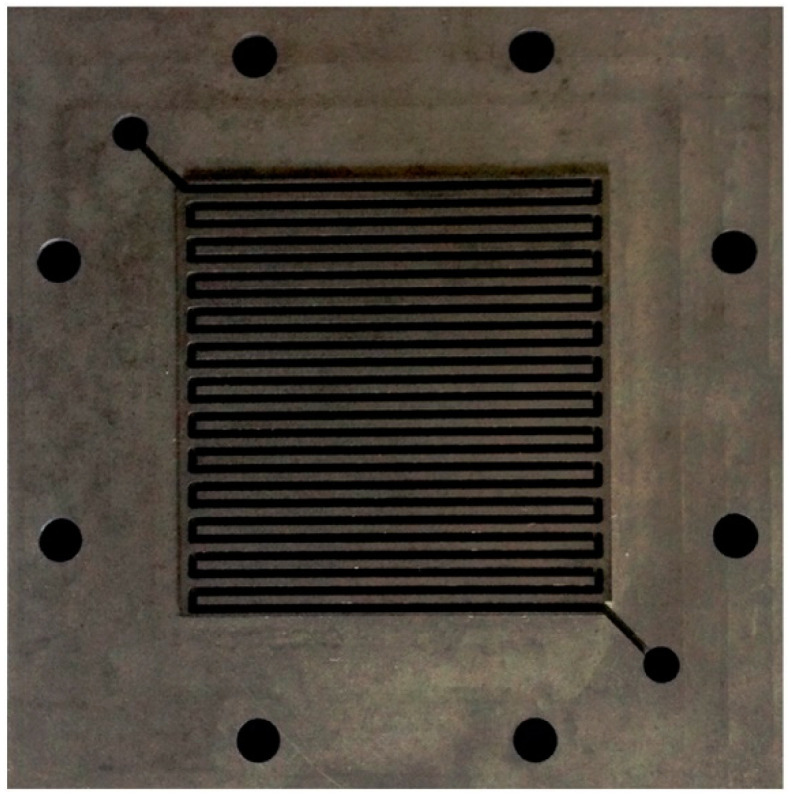
Serpentine flow fields. Reprinted from [26], copyright (2016), with permission from Elsevier.

**Figure 3 materials-14-02682-f003:**
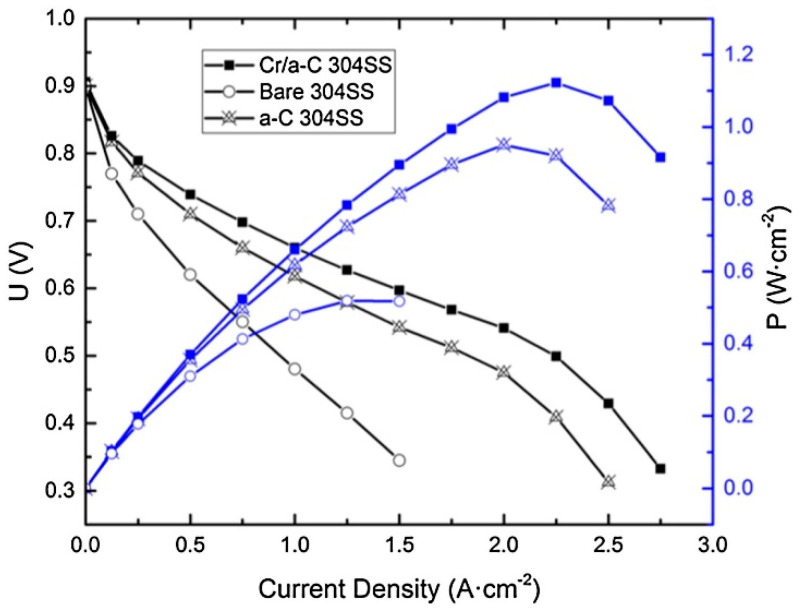
Outputs of assembled fuel cells with different bipolar plates. Reprinted from [26] copyright (2016), with permission from Elsevier.

**Figure 4 materials-14-02682-f004:**
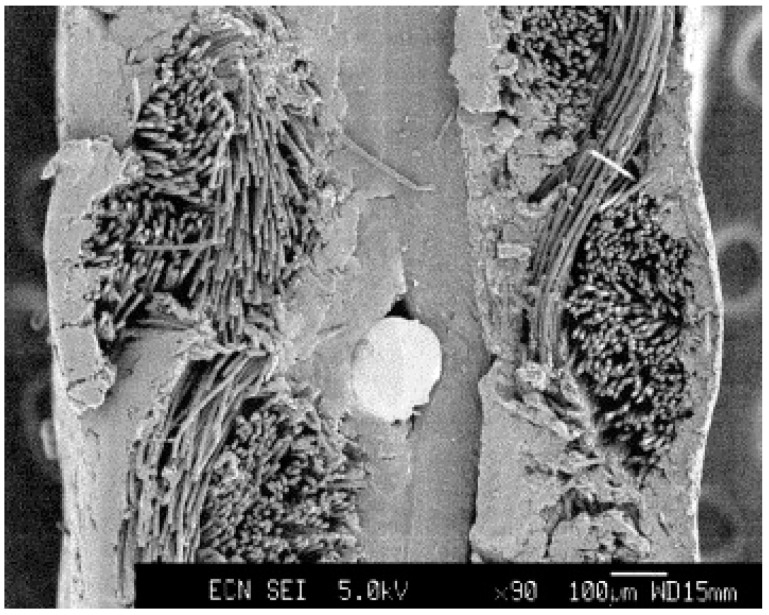
MEA cross section for in situ measurement according to Makkus (Au wire placed between membrane and E-tek backing plus electrode). Reprinted from [64] copyright (2000), with permission from Elsevier.

**Figure 5 materials-14-02682-f005:**
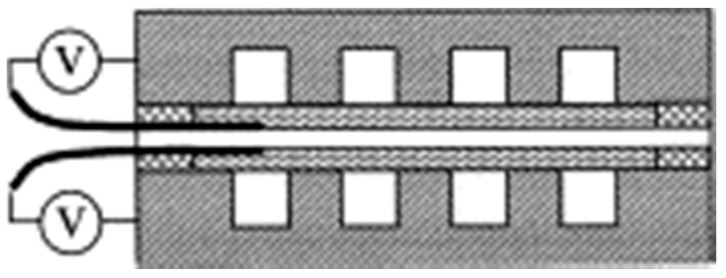
Scheme of method for in situ contact resistance measurement according to Makkus. Reprinted from [64] copyright (2000), with permission from Elsevier.

**Figure 6 materials-14-02682-f006:**
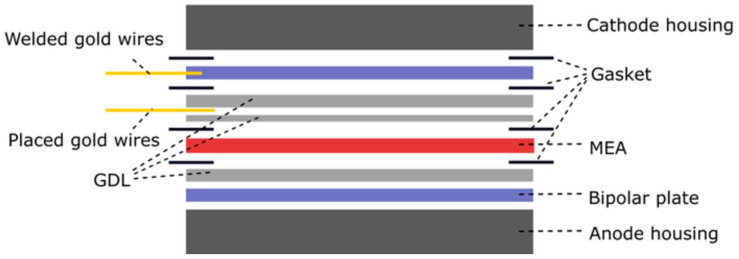
In situ contact resistance measurement according to Lædre [65].

**Figure 7 materials-14-02682-f007:**
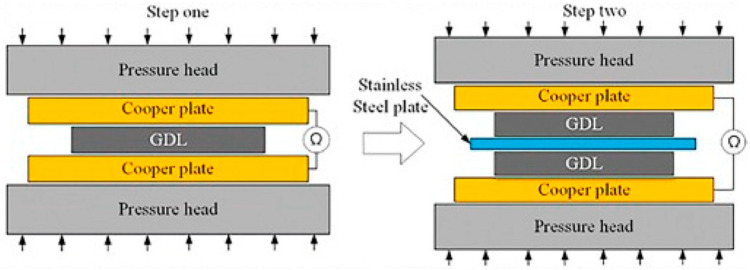
Principle of contact resistance measurement; stainless steel plate as a sample. Reprinted from [70] copyright (2018), with permission from Elsevier.

**Figure 8 materials-14-02682-f008:**
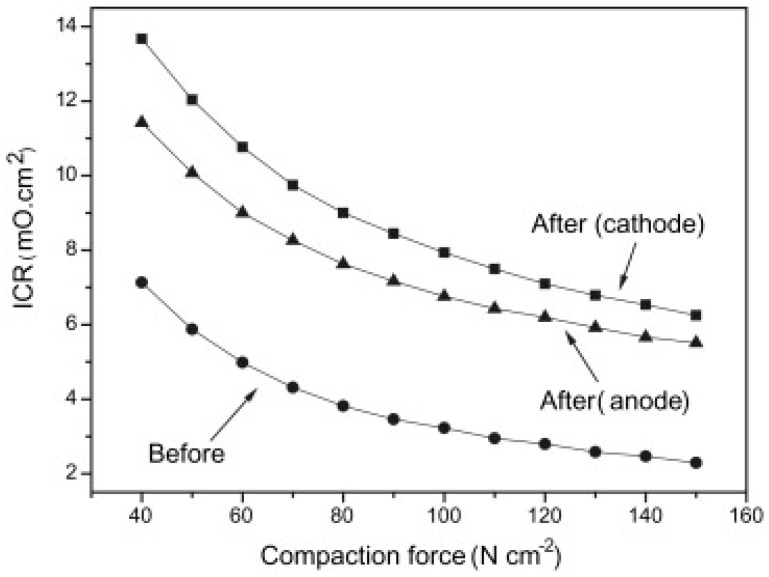
Contact resistance dependence on contact force. Reprinted from [45] copyright (2010), with permission from Elsevier.

**Figure 9 materials-14-02682-f009:**
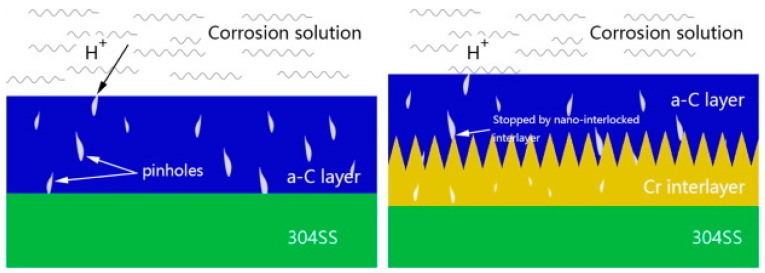
Principle of Cr-interlayer effect for amorphous carbon coating. Reprinted from [26] copyright (2016), with permission from Elsevier.

**Figure 10 materials-14-02682-f010:**
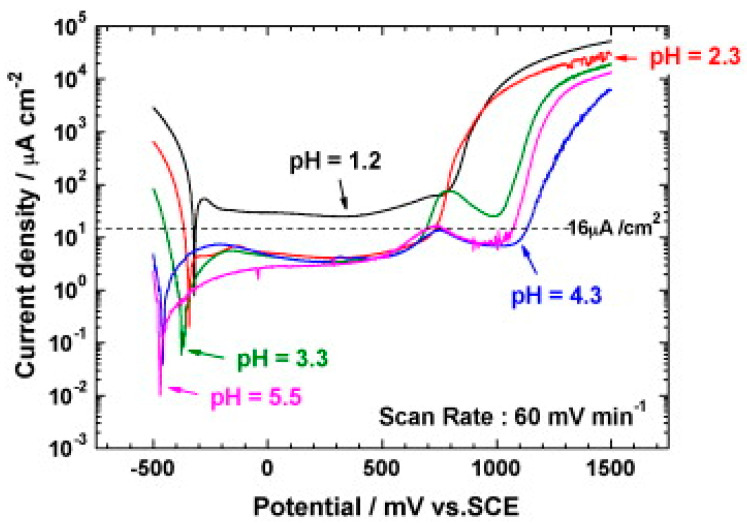
Effect of pH on the polarization curve of 310S stainless steel in 0.05 M H_2_SO_4_ with 2 ppm F^−^. Reprinted from [52] copyright (2008), with permission from Elsevier.

**Figure 11 materials-14-02682-f011:**
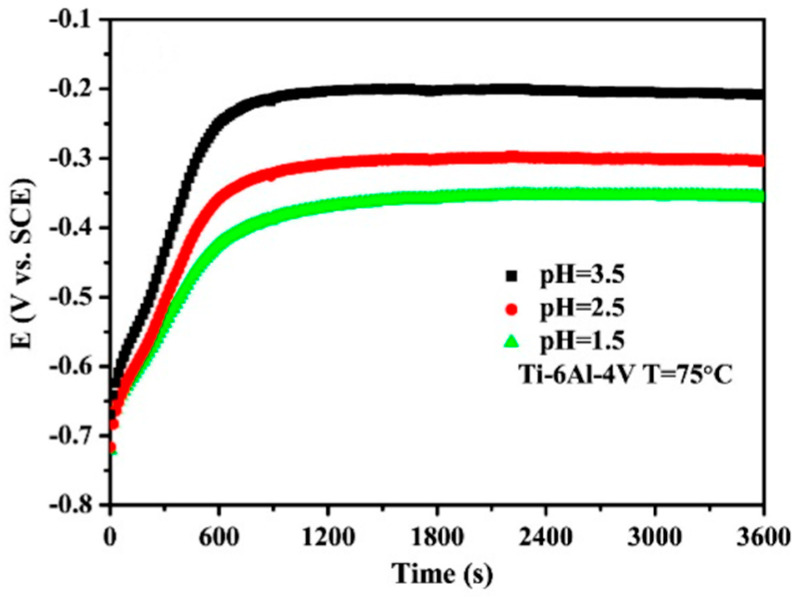
Influence of electrolyte pH (0.05 M SO_4_^2−^ with 2 ppm HF) on open-circuit potential of Ti-Al6-V4. Reprinted from [31] copyright (2016), with permission from Elsevier.

**Figure 12 materials-14-02682-f012:**
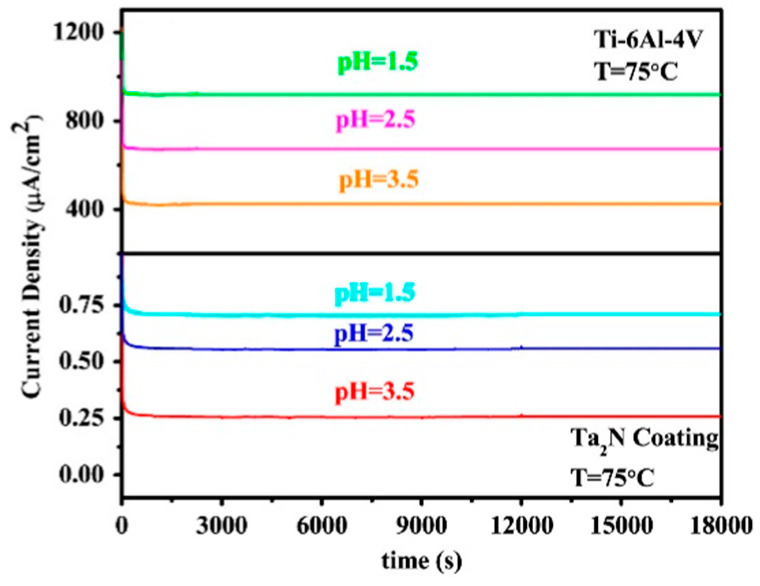
Influence of electrolyte pH (0.05 M SO_4_^2−^ with 2 ppm HF) on current density of Ti-Al6-V4 and Ta_2_N coating at potentiostatic polarization. Reprinted from [31] copyright (2016), with permission from Elsevier.

**Figure 13 materials-14-02682-f013:**
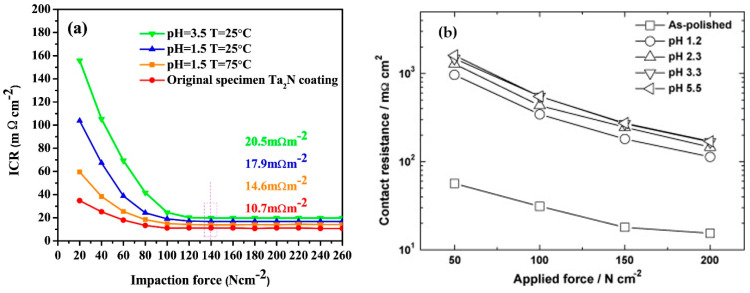
(**a**) Ta_2_N coated Ti-Al6-V4 alloy before and after PST test in 0.05 M H_2_SO_4_ with 2 ppm HF. Reprinted from [31] copyright (2016), with permission from Elsevier. (**b**) Effect of pH and temperature on contact resistance- 310S steel on 0.05 M H_2_SO_4_ with 2 ppm HF. Reprinted from [52] copyright (2008), with permission from Elsevier.

**Figure 14 materials-14-02682-f014:**
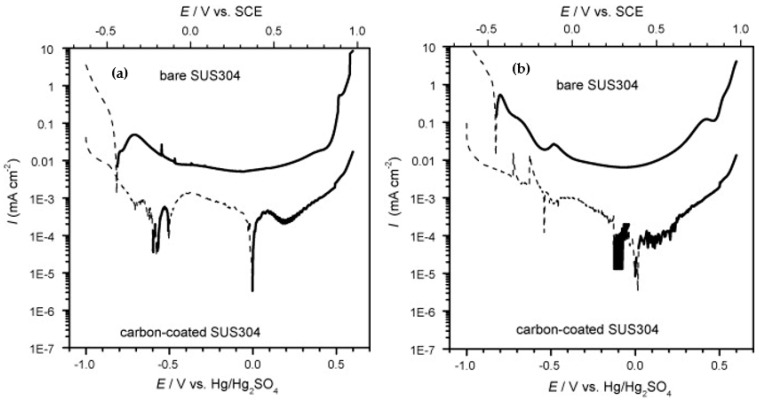
Comparison of corrosion behavior of steel in 0.5 M H_2_SO_4_ at 80 °C without (**a**) and with 2 ppm HF (**b**). Reprinted from [77] copyright (2007), with permission from Elsevier.

**Figure 15 materials-14-02682-f015:**
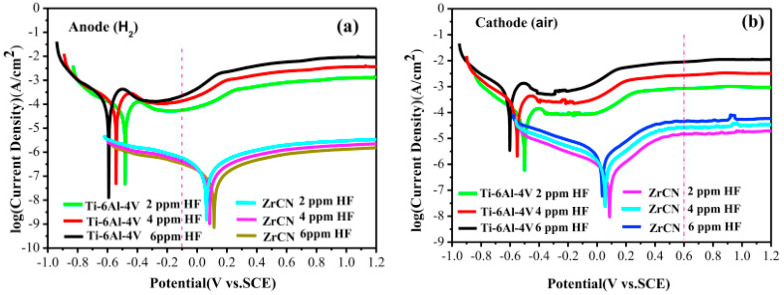
Influence of fluoride addition on corrosion behavior of Ti-Al6-V4 alloy and ZrCN coating. Reprinted from [66] copyright (2015), with permission from Elsevier.

**Figure 16 materials-14-02682-f016:**
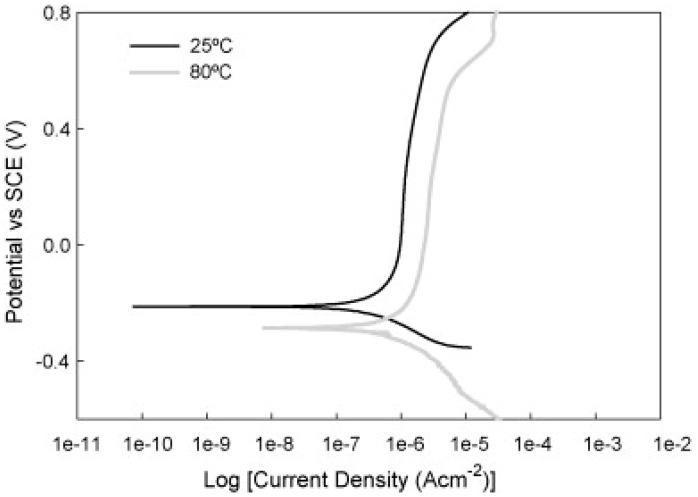
Comparison of SS 316 behavior at 25 °C and 80 °C in 0.5M sulfuric acid, pH = 4 and oxygen bubbled. Reprinted from [183] copyright (2007), with permission from Elsevier.

**Figure 17 materials-14-02682-f017:**
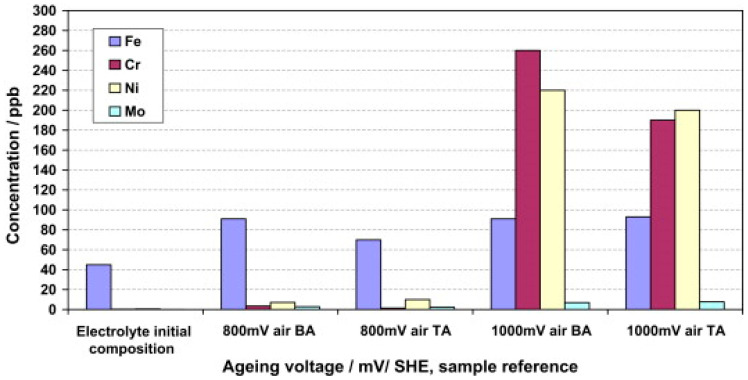
The amount of ions released after 500 h of exposure of AISI 316L depending on the applied potential (BA = bright annealing state). Reprinted from [16] copyright (2015), with permission from Elsevier.

**Figure 18 materials-14-02682-f018:**
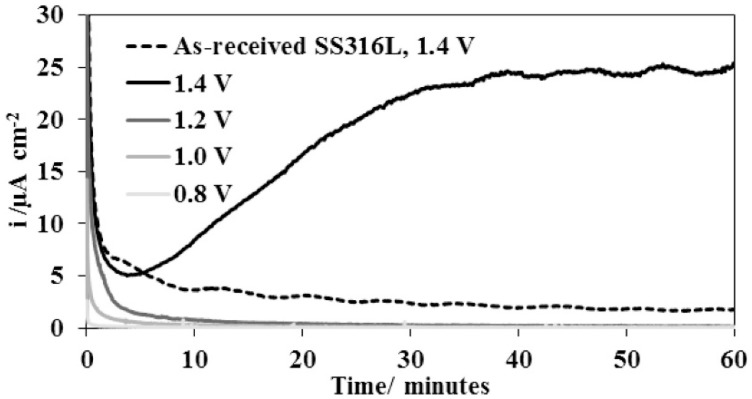
Current density dependence on applied potential for 316L steel with TiN coating in 1 mM H_2_SO_4_. Reprinted from [23] copyright (2010), with permission from Elsevier.

**Figure 19 materials-14-02682-f019:**
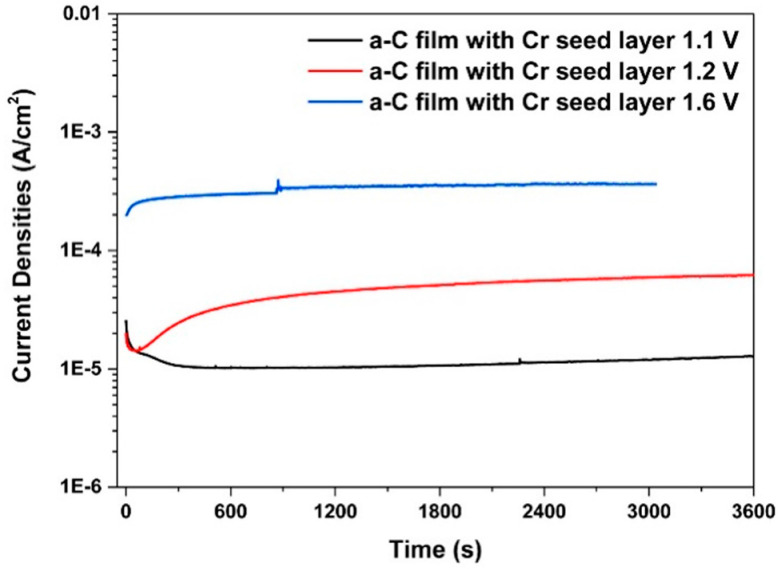
Effect of applied potential on current density of 316L steel coated with amorphous carbon with chromium interlayer. Reprinted from [24] copyright (2017), with permission from Elsevier.

**Figure 20 materials-14-02682-f020:**
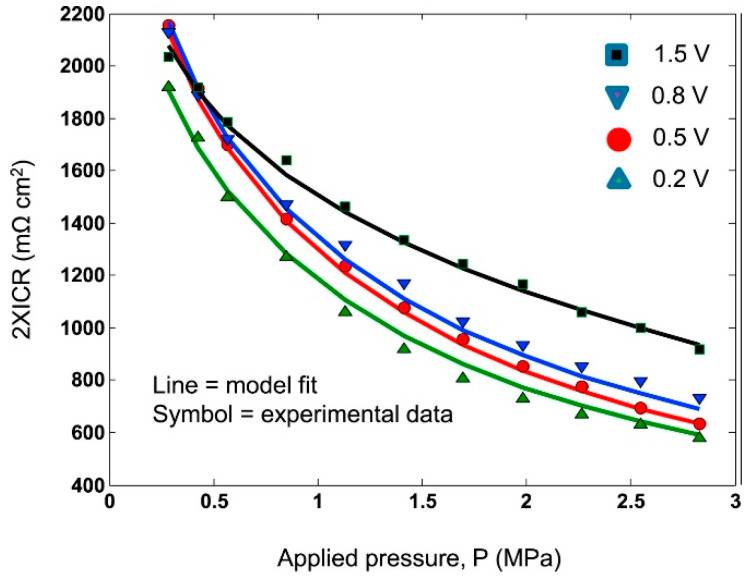
Potential impact on 316L steel contact resistance in sulfuric acid, pH = 3. Reprinted from [185] copyright (2014), with permission from Elsevier.

**Figure 21 materials-14-02682-f021:**
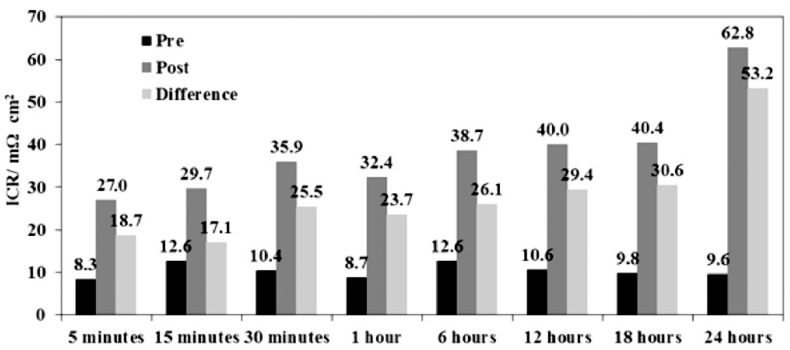
Effect of exposure time on contact resistance values for TiN coating on 316L steel before and after potentiostatic test in 1 mM H_2_SO_4_. Reprinted from [23] copyright (2015), with permission from Elsevier.

**Table 1 materials-14-02682-t001:** Requirements for bipolar plates for vehicles [11].

Monitored Property	Unit	2015 Status	Objectives 2020
Price	$/kW_t_	7 ^a^	3
Weight	kg/kW	<0.4	0.4
Hydrogen permeability coefficient ^1^	Std cm^3^/(s cm^2^Pa)	0	<1.3 × 10^−14^
Anode corrosion ^2^	µA/cm^2^	No active peak	<1 without active peak
Cathode corrosion ^3^	µA/cm^2^	<0.1	<1
Electrical conductivity	S/cm	>100 ^b^	>100
Areal specific resistance ^4^	ohm cm^2^	0.006 ^c^	<0.01
Flexural strength	MPa	>34	>25
Forming elongation	%	20–40	40

^1^ Measured at 80 °C, 3 atm, 100% RH. ^2^ pH = 3, 0.1 ppm HF, 80 °C, argon vented (potentiodynamic test: 0.1 mV/s, range −0.4 V to 0.6 V vs. ACLE). ^3^ pH = 3, 0.1 ppm HF, 80 °C, aerated solution (potentiostatic test at 0.6 V/ACLE, 24 h). ^4^ Includes interfacial contact resistance, measured at 138 N/cm^2^ before and after potentiostatic test. ^a^ Cost when producing sufficient plates for 500,000 systems per year. DOE Hydrogen and Fuel Cells. ^c^ Ref. [12]. ^b^ Annual Progress Report [13].

**Table 2 materials-14-02682-t002:** Chemical composition of tested steels according to ASTM.

AISI No.	DIN Equation	C	Si	Mn	P	S	Cr	Mo	Ni	Other
201	1.4372	≤0.15	≤1.00	5.5–7.5	≤0.060	≤0.030	16.0–18.0	-	3.5–5.5	N ≤ 0.25
254SMO	1.4547	≤0.02	≤0.80	≤1.00	≤0.030	≤0.010	19.5–20.5	6.0–6.5	17.5–18.5	Cu 0.50–1.00 N 0.18–0.22
304	1.4301	≤0.07	≤0.75	≤2.00	≤0.045	≤0.030	17.5–19.5	-	8.0–10.5	N ≤ 0.1
304L	1.4307	≤0.03	≤0.75	≤2.00	≤0.045	≤0.030	17.5–19.5	-	8.0–12.0	N ≤ 0.10
310	1.4845	≤0.25	≤1.50	≤2.00	≤0.045	≤0.030	24.0–26.0	-	19.0–22.0	-
310S	1.4845	≤0.08	≤1.50	≤2.00	≤0.045	≤0.030	24.0–26.0	-	19.0–22.0	-
316	1.4401 1.4436	≤0.08	≤0.75	≤2.00	≤0.045	≤0.030	16.0–18.0	2.00–3.00	10.0–14.0	N ≤ 0.10
316L	1.4404	≤0.03	≤0.75	≤2.00	≤0.045	≤0.030	16.0–18.0	2.00–3.00	10.0–14.0	N ≤ 0.10
321	1.4541	≤0.08	≤0.75	≤2.00	≤0.045	≤0.030	17.0–19.0	-	9.0–12.0	Ti 5xC-0.70 N ≤ 0.10
347	1.4550	≤0.08	≤0.75	≤2.00	≤0.045	≤0.030	17.0–19.0	-	9.0–13.0	Nb+Ta 10xC-1.0
436	1.4526	≤0.12	≤1.00	≤1.00	≤0.040	≤0.030	16.0–18.0	0.75–1.25	-	Nb+Ta 5xC-0.80
430	1.4016	≤0.12	≤1.00	≤1.00	≤0.040	≤0.030	16.0–18.0	-	-	-
446	1.4749	≤0.20	≤1.00	≤1.50	≤0.040	≤0.030	23.0–27.0	-	≤0.75	N ≤ 0.25
654SMO S32654	1.4652	≤0.02	≤0.50	2.0–4.0	≤0.030	≤0.005	24.0–25.0	7.0–8.0	21.0–23.0	Cu 0.30–0.60 N 0.45/0.55
904L	1.4539	≤0.02	≤1.00	≤2.00	≤0.045	≤0.035	19.0–23.0	4.0–5.0	23.0–28.0	Cu 1.0–2.0 N ≤ 0.10
S32205	1.4462	≤0.03	≤1.00	≤2.00	≤0.030	≤0.020	22.0–23.0	3.0–3.5	4.5–6.5	N 0.14–0.20

**Table 3 materials-14-02682-t003:** Comparison of materials at different temperatures in 85% H_3_PO_4_. Reprinted from [163] copyright (2010), with permission from Elsevier.

Material	Corrosion Rate [mm/a]		
	30 °C	80 °C	120 °C
AISI 316L	0.037	0.73	1.46
AISI 321	<0.01	0.12	0.46
AISI 347	<0.01	0.29	0.92
Inconel 625	<0.01	<0.01	0.23
Inconel 825	<0.01	0.23	0.37
Hastelloy C-275	<0.01	0.05	0.28

**Table 4 materials-14-02682-t004:** Dependence of polarization resistance (Rp) on exposure time [28]. Copyright Wiley-VCH GmbH. Reproduced with permission.

316L		904L		254SMO	
Time (h)	R_p_ (kΩ)	Time (h)	R_p_ (kΩ)	Time (h)	R_p_ (kΩ)
2	63	2	56	36	260
69	280	22	213	108	320
94	372	52	305	145	360
124	404	122	500	190	410
172	450	215	665	339	460
292	490	292	1400	425	480
340	484	358	1600	548	540

## Data Availability

No new data were created or analyzed in this study. Data sharing is not applicable to this article.

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
