# Peer review of "Metallic Material Selection and Prospective Surface Treatments for Proton Exchange Membrane Fuel Cell Bipolar Plates—A Review"

_materials, 2021, doi:10.3390/ma14102682_

Round 1
Reviewer 1 Report
In the manuscript titled "Metallic materials selection and prospective surface treatments 2 for fuel-cell bipolar plates- a review", the authors discussed the various metallic bipolar plate materials for fuel cell application. However, the manuscript should be significantly improved before publication in the journal "Materials". The authors are suggested to address the following comments to improve the manuscript quality.
- In the title, the fuel cell type should be specifically mentioned. The manuscript only discusses the metallic bipolar plate materials for the PEM fuel cell specifically.
- It seems that the article is lacking the most recent literature. The authors should include the most recent literature. Also, the literature from section "7. Latest progress in research" should be included in the relevant sections.
- Compared to the stainless steel materials sections the other sections such as Aluminum and Titanium are very poorly structured. Authors are suggested to improve those sections.
- Also, if possible, the addition of a section related to metallic bipolar plates for "Anion Exchange Membrane Fuel Cell" can improve the manuscript quality.
Author Response
The authors are grateful for helpful reviewer’s comments. We made our best to improve the manuscript. The answers to the particular comments are:
- In the title, the fuel cell type should be specifically mentioned. The manuscript only discusses the metallic bipolar plate materials for the PEM fuel cell specifically.
Title of the article was modified according to the reviewer’s recommendation. Now the title reflects the specific type of fuel-cell: Metallic materials selection and prospective surface treatments for proton exchange membrane fuel-cell bipolar plates- a review.
- It seems that the article is lacking the most recent literature. The authors should include the most recent literature. Also, the literature from section "7. Latest progress in research" should be included in the relevant sections.
Literature from section 7 was included into the relevant section and significant portion of recent literature was added. All major added parts are highlighted with yellow colour.
- Compared to the stainless steel materials sections the other sections such as Aluminum and Titanium are very poorly structured. Authors are suggested to improve those sections.
The review is mostly focused on stainless steel materials as it is believed that stainless steel is the most promising metallic material for bipolar plates considering corrosion resistance of a self-standing material (contrary to aluminum) and reasonable price (contrary to titanium). Other metallic materials were mentioned only for better understanding of the range of materials for bipolar plates.
- Also, if possible, the addition of a section related to metallic bipolar plates for "Anion Exchange Membrane Fuel Cell" can improve the manuscript quality.
This article is related to PEMFC and possible usage of various methods, techniques and materials to improve its performance, bipolar plates respectively. Anion exchange membrane fuel cells were not considered.
Reviewer 2 Report
The review article titled "Metallic materials selection and prospective surface treatments for fuel-cell bipolar plates" focuses on metallic bipolar plates, which benefit from many properties required for fuel-cells. Suitable coating systems prepared by a proper deposition method are presented, they make the replacement of graphite bipolar plates possible.
I recommend publication of this manuscript in the present form.
Author Response
The authors are grateful for reviewer’s recommendation. We appreciate that a lot.
Reviewer 3 Report
The research activities about metallic bipolar plates for fuel cells are well summarized in this review manuscript. Therefore, I could recommend this review for the potential publication in Materials. However, the following issues should be addressed before the publication.
1. Most importantly, there is no information on how this review is different from previous review papers covering the similar scope. The originality should be mentioned compared to the previous reviews in the beginning part.
2. I don't feel any necessity about Chap.7 (Lastest progress in research). Most papers covered in Chap.7 could be included in Chap.5.
3. Although this review appears to aim at covering the relevant references comprehensively, a couple of related references are missing.
- Park et al., "Passivation behavior and surface resistance of electrodeposited nickel-carbon composites", Electrochemistry, 82, 561 (2014)
- Park et al., "Corrosion prevention of chromium nitride coating with an application to bipolar plate materials", Electrochemistry, 82, 658 (2014)
4. The followings are minor points that could be corrected or revised.
- L26. Why not use proton exchange membrane fuel cell for PEMFC instead of polymer permeable membrane fuel cell?
- L66. once(?)
- Table 2.1: any meaning of superscript "b" and "h"?
- Fig. 4.7 is missing.
- L.380. uncouated(?)
- Chap. 6.1. Rather than the use of "he", a gender-neutral word would be appropriate. A space is needed between numbers and "M", for example, in 1M.
- L.1024. A parenthesis is missing.
- L.1034. Fig. 6.7 is not relevant.
Author Response
The authors are grateful for helpful reviewer’s recommendation. The particular comments are addressed below:
- Most importantly, there is no information on how this review is different from previous review papers covering the similar scope. The originality should be mentioned compared to the previous reviews in the beginning part.
Brief part was added to section 3. Information in abstract also show different approach to comparison of coatings and electrochemical results. We aimed in extracting important aspects for future studies. The abstract was provided with a sentence pointing at this original approach (yellow highlighted).
- I don't feel any necessity about Chap.7 (Lastest progress in research). Most papers covered in Chap.7 could be included in Chap.5.
Section 7 was incorporated into the relevant sections. Also, significant portion of recent literature was also added. All major added parts are yellow highlighted.
- Although this review appears to aim at covering the relevant references comprehensively, a couple of related references are missing.
- Park et al., "Passivation behavior and surface resistance of electrodeposited nickel-carbon composites", Electrochemistry, 82, 561 (2014)
- Park et al., "Corrosion prevention of chromium nitride coating with an application to bipolar plate materials", Electrochemistry, 82, 658 (2014)
Thank you for the references. More references including those two papers ([161,162]) have been involved.
- The followings are minor points that could be corrected or revised.
- L26. Why not use proton exchange membrane fuel cell for PEMFC instead of polymer permeable membrane fuel cell?
“Proton exchange membrane” reveals clearly the principle of the particular type of a fuel cell. Reflecting the comment of other reviewer, this specification has been involved in the title.
- L66. once(?)
Changed to “one”.
- Table 2.1: any meaning of superscript "b" and "h"?
The explanation has been added into the footnote of the table.
- Fig. 4.7 is missing.
The numbering has been revised and corrected.
- L.380. uncouated(?)
The misspelling has been removed.
- Chap. 6.1. Rather than the use of "he", a gender-neutral word would be appropriate. A space is needed between numbers and "M", for example, in 1M.
“He” was replaced with a gender-neutral expression “author”. Missing space has been inserted in all the cases.
- L.1024. A parenthesis is missing.
The error has been removed.
- L.1034. Fig. 6.7 is not relevant.
The link has been removed.
Round 2
Reviewer 1 Report
The review comments are addressed by the authors and the manuscript can be accepted for publication.